# *PiggyBac* transposon tools for recessive screening identify B-cell lymphoma drivers in mice

Julia Weber et al.[#]

B-cell lymphoma (BCL) is the most common hematologic malignancy. While sequencing studies gave insights into BCL genetics, identification of non-mutated cancer genes remains challenging. Here, we describe *PiggyBac* transposon tools and mouse models for recessive screening and show their application to study clonal B-cell lymphomagenesis. In a genome-wide screen, we discover BCL genes related to diverse molecular processes, including signaling, transcriptional regulation, chromatin regulation, or RNA metabolism. Cross-species analyses show the efficiency of the screen to pinpoint human cancer drivers altered by non-genetic mechanisms, including clinically relevant genes dysregulated epigenetically, transcriptionally, or post-transcriptionally in human BCL. We also describe a CRISPR/Cas9-based *in vivo* platform for BCL functional genomics, and validate discovered genes, such as *Rfx7*, a transcription factor, and *Phip*, a chromatin regulator, which suppress lymphomagenesis in mice. Our study gives comprehensive insights into the molecular landscapes of BCL and underlines the power of genome-scale screening to inform biology.

---

B-cell lymphoma (BCL) accounts for a considerable number of cancer cases[1], with more than 300,000 new patients per year worldwide[2]. Diffuse large B-cell lymphoma (DLBCL) is the most common form of BCL. Patients with this aggressive disease are often diagnosed at advanced stages[3]. Next-generation sequencing (NGS) studies have given comprehensive insights into BCL genetics[4–9] but also revealed that the molecular basis of the disease is still only partly understood. For example, it is still challenging to pinpoint drivers among the thousands of epigenetically, transcriptionally, or post-transcriptionally dysregulated genes in human BCL.

Insertional mutagenesis screens using retroviruses or DNA transposons can overcome some of these limitations of classic approaches to cancer genome analysis, particularly with regard to the analysis of the non-mutated cancer genome[10,11]. *Sleeping Beauty* (*SB*) and *PiggyBac* (*PB*) transposition systems originate from transposable elements in the genome of salmonid fish[12] and the cabbage looper moth *Trichoplusia ni*[13], respectively. Engineering of modified transposon versions allowed their application in mammalian systems, including germline manipulation[14–17] and insertional mutagenesis in mice[18–22]. Transposon mutagenesis has been extensively used in the hematopoietic system to discover novel cancer genes in various entities, including acute[23,24], and chronic[25,26] leukemia. However, only few data are available on lymphomagenesis[27]. Whole-body *PB* mutagenesis, for example, only rarely induces BCL[20] and could so far not be deployed for BCL screening. *SB* and *PB* are complementary tools with many different properties regarding cargo capacity, local hopping tendency, integration preferences, and other features[11,28]. As a consequence, screens performed with the two systems identify not only common but also many non-redundant genes[18–20,22,29].

Cytogenetic studies and retroviral insertional mutagenesis unraveled many of the key oncogenes driving B-cell lymphomagenesis[30,31]. Examples are *BCL2*, *BCL6*, and *MYC*, which have been extensively studied mechanistically[32]. In contrast, the role of tumor suppressors is generally less well understood. To address this need, we developed transposition systems for tumor suppressor screening in mice. Thereby, insertional mutagenesis is achieved by mobilization of promotor-less transposons carrying bidirectional gene trapping cassettes. We expected this system to be very efficient for the discovery of haploinsufficient tumor suppressor genes (TSG). However, insertions of transposons in both alleles of a gene is extremely unlikely to occur in the same cell, making recessive screening a challenge. In human cancer evolution, mutation in one allele of a tumor suppressor is often followed by loss of the wild-type allele, e.g., through deletion or mitotic recombination. To model and accelerate this process in mice, we took advantage of the hypomorphic *Bloom* allele (*Blm^{m3}*)[33]. The Bloom syndrome RecQ-like DNA helicase contributes to the maintenance of genomic integrity through its involvement in the homologous recombination pathway[34]. *Blm^{m3/m3}* mice display highly elevated LOH rates through sister chromatid exchange or copy number variation[33,35–37]. Hence, we aimed at exploiting this model for recessive screening in the context of transposon mutagenesis. Another limitation of whole-body transposon screens is that BCL phenotypes are only rarely induced. *Blm^{m3/m3}* mice are prone to B-cell lymphomagenesis[33], thus overcoming this problem.

Here, we combine the *Blm^{m3/m3}* allele with an inactivating *PB* transposon system in mice to achieve genome-wide TSG screening in BCL. We identify known and novel DLBCL genes, validate selected candidate genes through a CRISPR/Cas9-based functional approach and show the clinical relevance of our findings using large human DLBCL patient cohorts.

## Results

**Development of inactivating transposon systems in mice.** A critical parameter affecting the success of TSG screens is the efficiency of gene inactivation. Intragenic transposon insertions are typically located in introns, which are much larger than exons. To achieve gene inactivation from intronic positions, transposons have to be designed to carry gene trapping elements. We first thoroughly tested different widely used splice acceptors (SA) at the *Hprt* locus. Efficient gene trapping at this X-chromosomal locus confers 6-thioguanine (6TG) resistance in mouse embryonic stem (ES) cells derived from male mice. Using recombinase-mediated cassette exchange, we shuttled different transposon variants carrying the adenovirus-derived SA (Av-SA), the *Engrailed-2* exon-2 SA (En2-SA), and the carp *β-actin* SA (Cβa-SA) to the *Hprt* locus and selected cells for 6TG resistance (Supplementary Figure 1). Trapping efficiencies were quantified by counting 6TG-resistant colonies and were highest for the Av-SA and the En2-SA.

Based on these results, we designed two transposon variants (*ITP1* and *ITP2*) that can inactivate genes independently of their orientation (Fig. 1a). Both transposons contain *PB* and *SB* inverted terminal repeats (ITR), allowing mobilization by either transposase. Between the ITRs, they harbor bidirectional poly-adenylation signals (pA), which are flanked by the Adv-SA and En2-SA. Additionally, *ITP1* contains a bGEO (β-galactosidase expression and neomycin resistance) reporter gene, which enables visualization of gene-trapping events. We used these constructs to generate five different transgenic transposon mouse lines, which differ in the location of the transposon concatemer and its size (2–70 transposon copies) (Fig. 1b). For subsequent experiments, we selected the *ITP1-C* and *ITP2-M* lines, which we intercrossed with *iPBase* knock-in mice (*Rosa26^{PB/+}*, expressing the insect version of the *PB* transposase constitutively; Fig. 1c), and *Blm^{m3/m3}* mice (Fig. 1c). We observed pronounced embryonic lethality in *ITP1-C;Rosa26^{PB/+};Blm^{m3/m3}* mice, with only 6.0% of the expected triple-transgenic mice being born. In contrast, *ITP2-M;Rosa26^{PB/+};Blm^{m3/m3}* mice were born in proportions closer to the calculated Mendelian frequency (45.7%) (Supplementary Data 1). These variations in embryonic lethality are most likely due to the different transposon copy numbers of the *ITP1-C* (70 copies) and *ITP2-M* (35 copies) lines.

***IPB* mice predominantly develop BCL.** We used the *ITP2-M* line to establish the screening cohorts, consisting of 123 experimental *ITP2-M;Rosa26^{PB/+};Blm^{m3/m3}* mice (hereafter referred to as *IPB*) and 87 *Rosa26^{PB/+};Blm^{m3/m3}* or *ITP2-M;Blm^{m3/m3}* control mice (for tumor spectrum see Supplementary Table 1). Animals were aged and monitored for tumor development. We observed a broad spectrum of cancer phenotypes in both cohorts, but tumor latency and survival was reduced substantially in *IPB* mice (Fig. 1d, e). We collected tumors from 82 animals and characterized them histopathologically. Roughly two-thirds of tumors were hematopoietic neoplasms (n = 65), whereas 35% constituted a broad range of solid cancers (n = 35; Supplementary Figure 2; Supplementary Table 2). We characterized 59 hematopoietic tumors using an immunohistochemistry (IHC) panel comprising the markers B220 (specific for B cells), CD3 (T cells), myeloperoxidase (myeloid cells), and CD138 (plasma cells). The vast majority of tumors were B cell neoplasms (52/59; 88.1%). Only six CD3 positive T-cell lymphomas (10.2%) and one tumor with myeloid differentiation (1.7%) were found.

**Histopathological and molecular characterization of BCLs.** BCLs were almost exclusively reminiscent of human DLBCL (51/52; 98.1%; Fig. 2a; Supplementary Data 2). Neoplasms usually

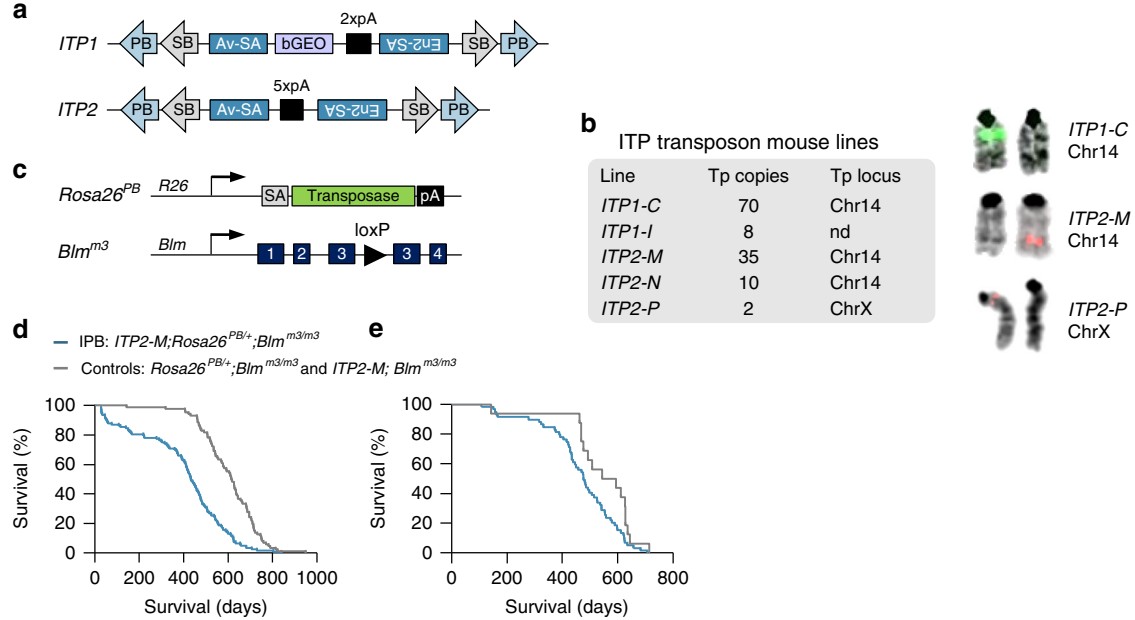

**Fig. 1** A *PiggyBac* transposon system for recessive screening in mice. **a** Structure of "inactivating transposons" *ITP1* and *ITP2*. **b** Transgenic mouse lines generated using *ITP1* or *ITP2*. Transposon copy numbers within the transgene array as well as the chromosomal donor locus of the array are shown. Chromosomal donor loci for different lines are shown on the right, as detected by fluorescence in situ hybridization with transposon-specific probes. Note that the *ITP2-M* and *ITP2-N* mouse lines emerged from a single founder animal. **c** Structures of the *Rosa26^PB^* and *Blm^m3^* alleles as described earlier[20,33]. The *Rosa26^PB^* knock-in allele expresses the insect version of the *PiggyBac* transposase constitutively driven by the endogenous *Rosa26* promoter. **d**, **e** Kaplan–Meier plots showing survival of *IPB* and control mice. In **d** the whole cohort is shown ($n = 123$ [IPB], $n = 87$ [controls]; $p < 0.0001$, log-rank test). **e** represents survival curves for mice displaying hematopoietic tumors characterized histopathologically ($n = 59$ [IPB], $n = 16$ [controls]; $p = 0.025$, log-rank test). PB *PiggyBac*, SB *Sleeping Beauty*, Av-SA adenovirus-derived splice acceptor, bGEO β-galactosidase/neomycin resistance reporter including the bovine growth hormone polyadenylation signal, En2-SA *Engrailed-2* exon-2 splice acceptor, pA SV40 bidirectional polyadenylation signal, Tp transposon, R26 *Rosa26*, SA splice acceptor, Blm *Bloom syndrome RecQ like helicase*, nd not done

manifested in mesenteric lymph nodes and/or spleens, with moderate or extensive alterations of lymphoid organ architecture due to abnormal B cell expansion (demonstrated using B220 IHC). DLBCLs were composed of large-sized neoplastic cells (centroblasts) with abundant cytoplasm, a round nucleus, vesicular chromatin, with two or more nucleoli, which were often membrane-bound, and showed high proliferation rates (as demonstrated by Ki-67 IHC).

In all tumors, we also observed a small percentage of lymphoid cells with immunoblastic morphology (larger cells with abundant cytoplasm and a single prominent centrally localized nucleolus).

A subset of DLBCL cases showed characteristics of plasmacytic differentiation (13/51; 25.5%), in which a fraction of tumor cells lost the B cell marker (B220) and expressed the plasma cell marker (CD138) (Supplementary Figure 3). A significant proportion of tumor cells, however, retained B220 expression, distinguishing these cancers from plasmablastic lymphoma or other plasma cell malignancies. In general, tumor cell infiltration into organs located within the thoracic and abdominal cavities, such as lungs, liver, intestine, and kidneys was frequently observed (37/42 analyzed DLBCLs; 88.1%; Supplementary Data 2) while bone marrow infiltration was rare (2/42; 4.8%; Supplementary Figure 4).

To sub-classify the DLBCL cases based on their cell of origin, we performed IHC using the germinal center marker Bcl6 and the post-germinal center marker Mum1/Irf4. Expression of MUM1/IRF4 is associated with non-germinal center B-cell like (GCB) DLBCL in humans[38]. In contrast, BCL6 expression is primarily associated with GCB DLBCL, although a subset of non-GCB DLBCL is also positive for BCL6[39]. We analyzed 20 samples and found that 15 cases (75%) were Bcl6-positive/Irf4-negative,

suggesting a GCB DLBCL phenotype. Five cases (25%) were Irf4-positive/Bcl6-negative, which we classified as non-GCB DLBCL (see representative images in panels 4 and 5 of Fig. 2a).

In addition, we performed RNA sequencing of DLBCL samples ($n = 25$) for gene-expression profiling (GEP), which is considered the gold standard for DLBCL sub-classification in humans[39] (Fig. 2b). Using the murine orthologues of the human classifier genes[40], mouse DLBCLs fell into two main clusters. Cluster B contained exclusively IHC-diagnosed GCB tumors, whereas all five IHC-diagnosed non-GCB cancers fell into cluster A (Fig. 2b). As in human DLBCL, IHC-based and GEP-based tumor classification are not fully superimposable[41]. In fact, the discordance in mice might be even stronger because mouse DLBCLs are less homogeneous: we observed that GCB tumors often contain infiltrates of CD3-positive T lymphocytes and residual plasma cells, which makes accurate GEP from whole tumor lysates challenging. This might be a reason why some of the IHC-diagnosed GCB samples fall into cluster A. Taken together, these data show that mouse DLBCLs can be sub-classified similarly to human DLBCLs. Thus, our model recapitulates key aspects of the human disease.

**B-cell receptor repertoire profiling of DLBCLs.** For clonality analysis, we performed RNA-based immunoglobulin repertoire profiling of 30 DLBCL cases (Fig. 3a; Supplementary Data 3). To this end, we conducted full-length amplification of the variable regions of the immunoglobulin heavy (IGH) and immunoglobulin light (IGL) chains. To eliminate PCR and sequencing errors leading to incorrect clone assignments, unique molecular identifiers (UMI) were introduced during cDNA synthesis. Immune

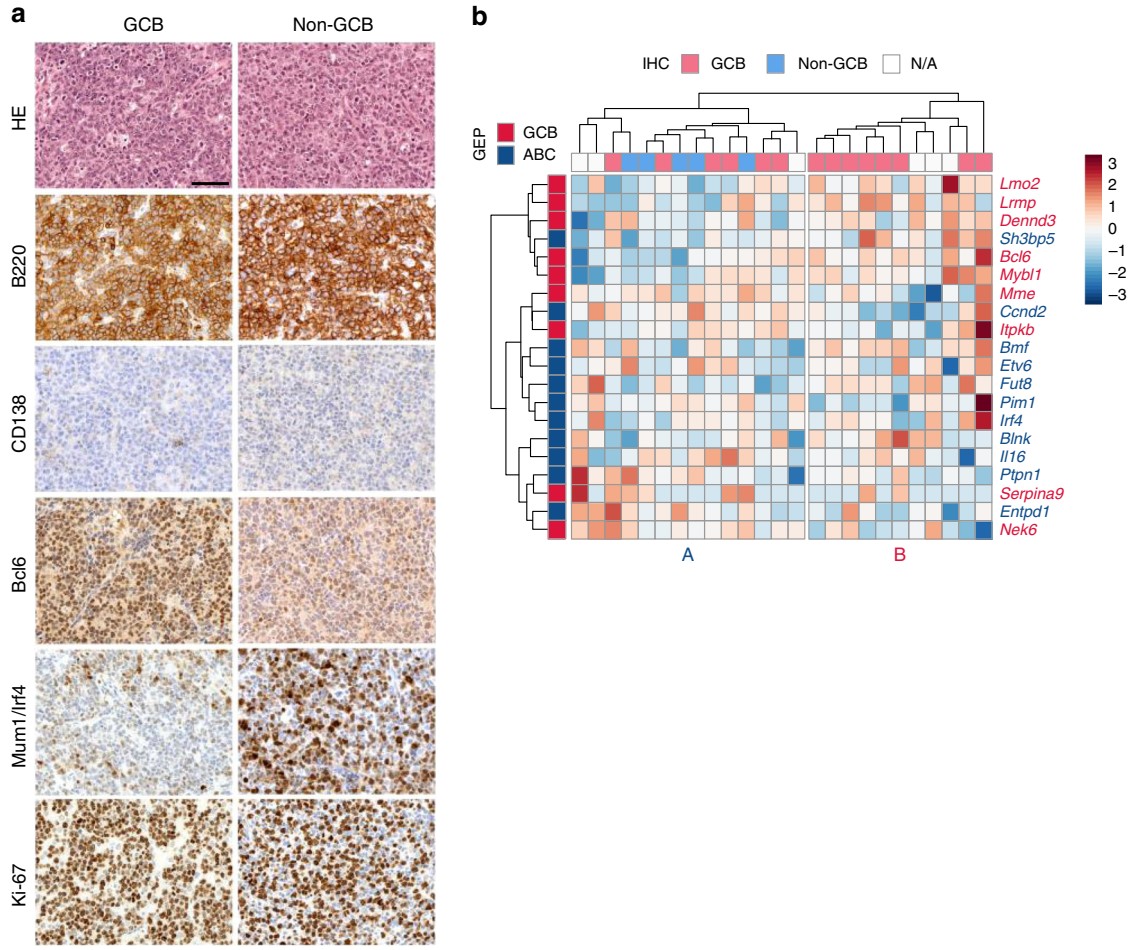

**Fig. 2** Characterization and classification of DLBCLs from *IPB* mice. **a** Immunohistochemical characterization and sub-classification of mouse DLBCLs ($n = 25$) from *ITP2-M;Rosa26^PB/+;Blm^m3/m3* (IPB) mice. Microscopic images of two representative DLBCL cases, one classified as germinal center B-cell like (GCB) DLBCL (left panel) and the other as non-GCB DLBCL (right panel). Tumors consist of large-sized neoplastic cells, show strong expression of B220/ CD45R (B-cell marker), and are negative for CD138 (plasma cell marker). While the GCB DLBCL sample exhibits high expression of Bcl6 and is negative for Mum1/Irf4, the non-GCB DLBCL case shows the opposite expression pattern. Both tumors show high proliferation rates as indicated by Ki-67 immunohistochemistry (IHC). Scale bar, 50 µm. **b** RNA sequencing-based sub-classification of DLBCLs ($n = 25$) from *IPB* mice. For clustering of the expression data, mouse orthologues to human DLBCL classifier genes[40] were used. The heatmap shows z-transformed expression values. Two main clusters (A and B) were identified. IHC-based subtyping (GCB/non-GCB) of the corresponding tumors is indicated at the top of the heatmap. GEP gene expression profiling, ABC activated B-cell like

repertoires were sequenced on the Illumina MiSeq. Data analysis, de-multiplexing and UMI consensus sequence assembly was performed with *MIGEC*[42]. *MiXCR* was used for clone detection based on the highly variable complementarity-determining region 3 (CDR3)[43]. To visualize the clonal structure of tumors, we developed *CloNet*, a pipeline for generation of clonality network plots (Fig. 3a, b, Supplementary Figures 5, 6, 7 and 8). In these plots, each clone constitutes a node of the network. The size of the node correlates with the number of reads assigned to it, and clones differing by only 1 bp in their CDR3 sequence are linked by edges. The complexity of the branching (i.e. number of sub-clones) is a measure for the grade of somatic hypermutation (SHM), which is a hallmark of DLBCL[44]. We highlighted clones that consist of more than 10% of the total reads of a sample in color (red, blue, or green). Different clones (as defined by a unique V(D)J rearrangement) are marked with different colors. Note that RNA was isolated from whole tumor tissue lysates that contain varying amounts of non-transformed B cells (most likely accounting for the small gray nodes in the plots). Figure 3c shows the proportion of DLBCL cases with different grades of

clonality. The vast majority of tumors (16/26) were monoclonal, indicating that these were full-blown malignant lymphomas arising from one transformed B cell. Eight samples consisted of two dominant clones, suggesting the presence of two independent malignant DLBCLs in one mouse, and only two tumors arose from multiple unrelated clones. We found evidence of SHM in the majority of tumors (Fig. 3b). As expected, there were differences in the extent of SHM between individual tumors, and between heavy and light chains of the same clone, as these undergo SHM separately.

**Characterization of DLBCLs by copy number analysis**. To further characterize DLBCLs, we carried out array comparative genomic hybridization, which revealed recurrent amplifications on chromosome 11qA1-B1.3 (10/16 samples; 62.5%; Fig. 4a, b). The 5.6 Mb large minimal overlap region (chr11:21888515-27523873) contains 27 protein-coding genes, including two well-known DLBCL oncogenes *Bcl11a* and *Rel*, as well as other genes with predicted oncogenic function, like *Commd1* and *Otx1*[45,46].

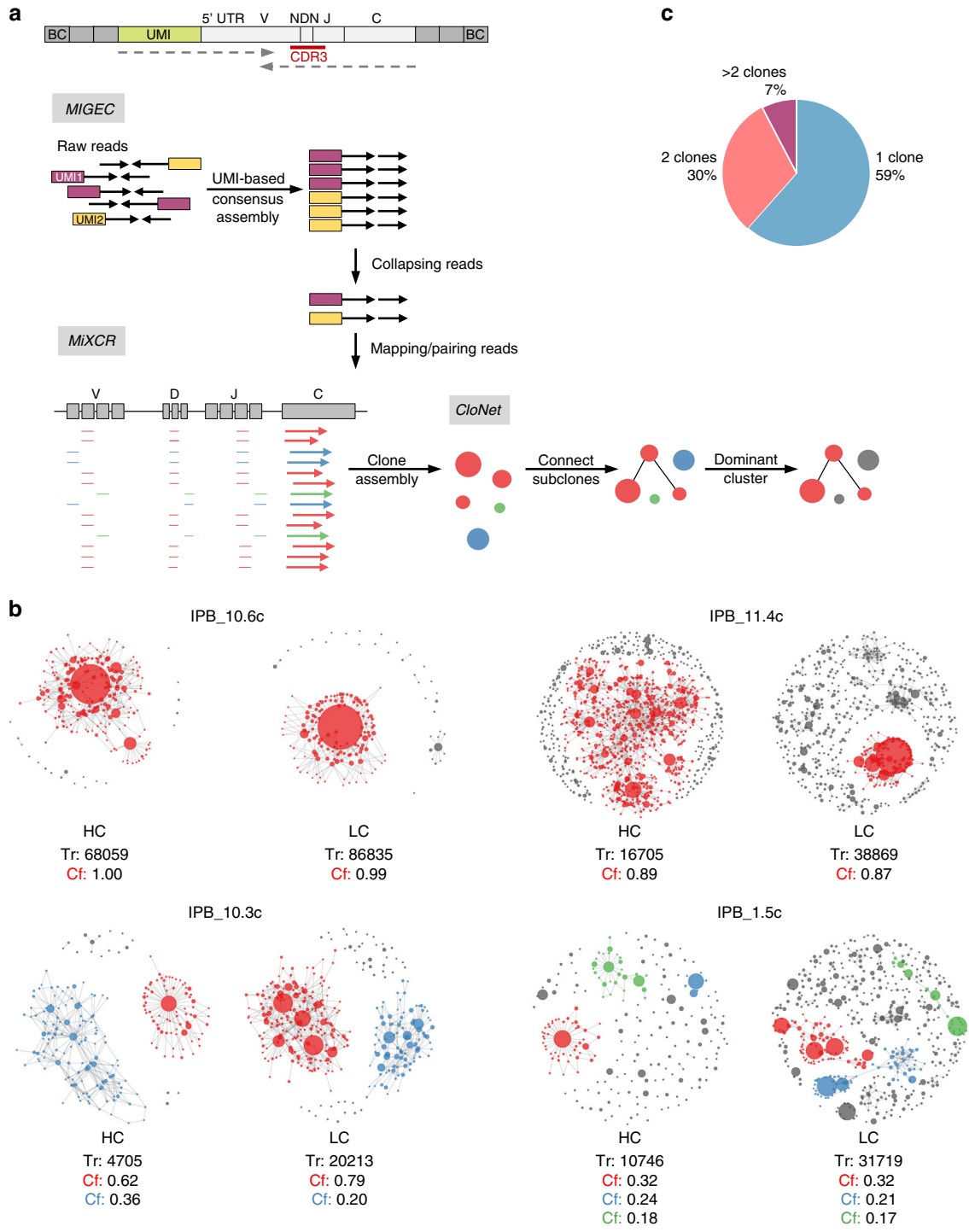

The concordant region in the human genome, 2p15-16, is frequently amplified in human DLBCL cases[47], demonstrating again the relevance of our model, which recapitulates one of the characteristic oncogenic events in human B-cell lymphomagenesis. Nevertheless, in contrast to human DLBCLs, which often show multiple copy number alterations[48], recurrently amplified/deleted regions in the murine DLBCL cases were infrequent. This suggests that transposon-driven mutagenesis is the key cancer-promoting factor driving lymphomagenesis, removing the need for such events, as also shown in previous transposon mutagenesis studies for osteosarcoma[49].

**QiSeq identifies cancer genes in DLBCL.** We performed quantitative transposon insertion site sequencing (QiSeq)[28] in 42 DLBCL cases (Fig. 4a and Supplementary Data 2) and recovered a total of 298,439 non-redundant insertions (supported by a minimal read coverage of 2, which was set as threshold for calling). Global analysis of insertion density/direction profiles showed no preference for a specific transposon orientation, which is in accordance with *ITP2*'s bidirectional gene inactivating capability (Fig. 4c). To detect genomic regions that are more frequently hit by the *ITP2* transposon than expected by chance, we applied CIMPL[50] (Common Insertion site Mapping Platform;

**Fig. 3** Clonality analysis of DLBCLs by immune repertoire sequencing. **a** Workflow for B-cell receptor repertoire analysis. Full-length amplification and sequencing of immunoglobulin heavy and light chain variable regions was performed from bulk tumor tissue ($n = 30$) of *ITP2-M;Rosa26^PB/+;Blm^m3/m3* (IPB) mice. Top image shows exemplary cDNA product after library preparation with amplified variable and constant region of the immunoglobulin heavy chain (light gray). Unique molecular identifiers (UMI; green) and adapters and barcodes for sequencing (dark gray) were introduced during library preparation. Dotted arrows indicate reads for 300 bp paired-end sequencing. Sequenced raw reads were de-multiplexed and a consensus read sequence for each UMI was assembled with *MIGEC*. All reads containing an identical UMI were collapsed into one read. *MiXCR* was used for mapping of reads to mouse reference sequences and clonotype assembly based on the complementarity-determining region 3 (CDR3) region. To visualize the clonal structure of individual tumors, we developed *CloNet*, a pipeline for generation of clonality network plots. **b** Exemplary clonality network plots derived from four different mouse DLBCL samples. Plots display clonal structures of immunoglobulin heavy and light chains. Each clone (defined by a unique CDR3 sequence) constitutes a node of the clonality network. The size of the node scales with the third root of the count of the reads assigned to it. A link between two nodes was drawn if the clones mapped to identical V and J genes and differed by at most 1 bp in their CDR3 sequence. The complexity of the branching of a clone (i.e. number of subclones) is a measure for the grade of somatic hypermutation. Clones defined by a unique V(D)J rearrangement that contained more than 10% of the total reads are highlighted in color. Two monoclonal (*IPB_10.6c and IPB_11.4c*) samples, one biclonal (*IPB_10.3c*) and one oligoclonal (*IPB_1.5c*) sample are shown. **c** Proportion of monoclonal (1 clone), biclonal (2 clones), and oligo-/polyclonal (>2 clones) DLBCL samples. BC barcode, V variable gene segment, NDN diversity gene segment, J joining gene segment, C constant gene segment, HC heavy chain, LC light chain, Tr total reads, Cf fraction of clone

based on a Gaussian kernel convolution framework), as well as TAPDANCE[51] (Transposon Annotation Poisson Distribution Association Network Connectivity Environment; Supplementary Data 4) analyses. The semi-quantitative nature of QiSeq allowed us to set read-coverage-based thresholds for CIMPL analysis, which identified 184 common insertion sites (CIS) when analyzing insertions with a read coverage $\geq 20$ ($n = 43,474$ non-redundant insertions; Supplementary Data 5; for co-occurrence analysis see Supplementary Table 3). Transposon insertions in CIS genes had typically no orientation bias and were distributed over the whole length of the genes. Examples for such gene inactivating insertion patterns, which predict a tumor suppressive function for target genes, are shown in Supplementary Figure 9. A few exceptions involving oncogene activation (e.g., by 3′ UTR interference or cryptic activation) are displayed in Supplementary Figure 10 for *Malt1* and *Rel*.

Figure 4d shows the top 50 CIS genes identified in the screen. Among them, 22 are either found in the Cancer Gene Census database and/or have known roles in human DLBCL, while 28 genes have not yet been linked to B-cell lymphomagenesis. Examples for the former include *Pten*, a major negative regulator of the Pi3k/Akt/Mtor signaling pathway, which mediates signals downstream of the B-cell receptor, and *Gna13*, for which inactivating mutations have recently been discovered in follicular lymphoma and DLBCL[52,53].

We next explored globally possible functions linked to our top 50 CIS genes. As expected, Reactome gene set enrichment analysis (using the top 50 genes as input) revealed enrichment of signatures typical for DLBCL, including "immune system" and "signaling mediated by the B-cell receptor" (Supplementary Data 6). To gain deeper insights into known or predicted/suspected molecular functions of the top 50 genes, we performed systematic literature search. As indicated in Fig. 4d, we identified genes with diverse molecular functions, including chromatin and transcriptional regulators (22/50), signaling mediators (13/50), and regulators of RNA metabolism (7/50).

We compared these CISs to the top 150 driver genes identified in the largest available human DLBCL sequencing study ($n = 1001$ cases)[4]. We found that while some CIS genes (13/50; 26%) are recurrently mutated (point mutations, indels) in the human DLBCL dataset, the vast majority of genes (37/50; 74%) is not. We therefore hypothesized that these unidentified genes are (i) not recurrently mutated, (ii) part of the non-mutated DLBCL genome altered by mechanisms other than mutation, such as transcriptional or epigenetic dysregulation, or (iii) large deletions/amplifications harboring hitherto unknown cancer genes. To examine this possibility, we compared expression of genes in DLBCL samples relative to non-malignant B cells (centroblasts)

(Fig. 5 and Supplementary Figure 11). Notably, human orthologues of mouse CIS genes were highly significantly enriched among the genes downregulated in human DLBCL ($p = 5.7 \times 10^{-16}$, Fisher's exact test; Fig. 5). In addition, analysis of copy number variation in human DLBCLs revealed almost exclusively deletions at loci harboring human orthologues of mouse CIS genes (Supplementary Figure 12). These cross-species comparisons provide further support for the power of the screen to identify tumor suppressors regulated in human DLBCL.

**Loss of heterozygosity in tumors originating from *IPB* mice.** The necessity to inactivate TSGs biallelically poses a major challenge for recessive screening approaches. In transposon mutagenesis screens, the likelihood of both alleles of a TSG being independently hit by two transposons in the same cell is extremely small. To overcome this problem, we performed the screen in a *Blm^m3/m3* background, which we hypothesized to enhance LOH rates at CIS loci. To examine whether LOH is indeed increased at TSGs hit by transposons, we analyzed the locus of *Pten*, which is one of the top CIS genes in the screen. One important consideration is that insertions can be subclonal, and in such cases bulk-sequencing-based LOH studies are not possible. To address this issue, we exploited the semi-quantitative nature of our insertion site sequencing approach (QiSeq), which allowed us to draw conclusions about the clonal representation of any given transposon integration. For LOH studies, we only used cancers with high-coverage *Pten* insertions, suggesting their presence in the major tumor clone. Ten such samples were available. None of them displayed focal *Pten* deletions, and only one high-coverage *Pten* insertion was observed per sample, suggesting that transposon-based *Pten* inactivation is mono-allelic.

To examine the presence or absence of LOH, we first performed single nucleotide polymorphism (SNP) analysis using amplicon-based NGS of the 10 DLBCL tissues and corresponding tail samples from *IPB* mice (Fig. 6a). In six animals, a heterozygous *Pten* germline variant was identified in the tail, allowing SNP-based LOH profiling at this locus. In three out of these six DLBCL cases, the heterozygous SNP was detected at similar variant frequencies in DLBCL tissue and tail, indicating either a lack of LOH in the tumor sample or a false-negative result due to contaminating non-tumor cells in the tissue (Fig. 6b). In the other three DLBCLs, variant frequencies deviated substantially from 0.5, clearly demonstrating LOH at the *Pten* locus (Fig. 6b). Thus, LOH occurred in at least 50% of those tumors in which LOH analysis was possible. We also observed LOH at other loci, such as the *Apc* gene in an *IPB* small intestine tumor (Supplementary Figure 13), showing that LOH is not restricted to the *Pten* locus.

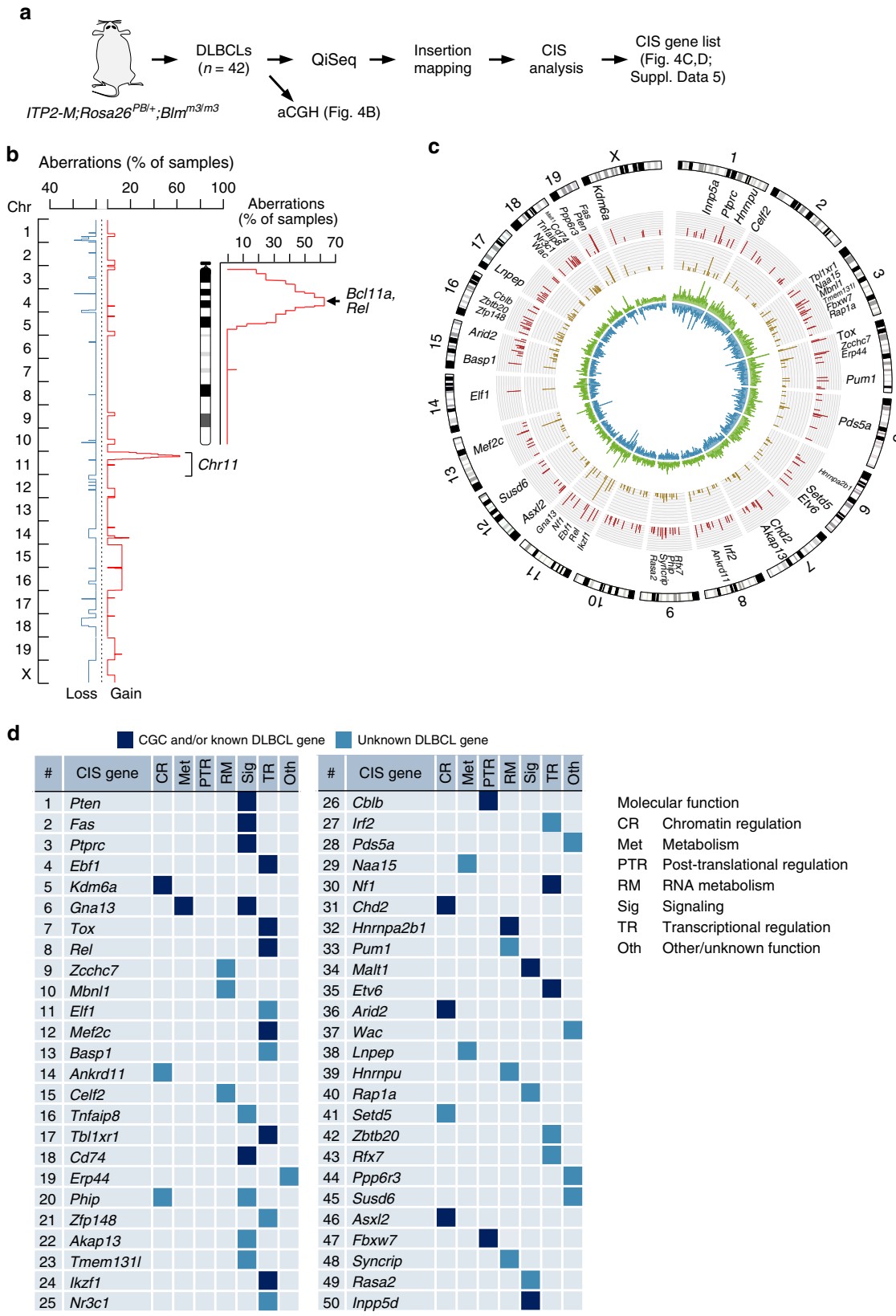

To examine LOH in samples where SNP-based LOH analysis was not possible (no heterozygous germline variants at the locus of interest), we also scored Pten expression by IHC in the 10 *IPB* DLBCLs with high-coverage *Pten* insertions (Fig. 6c). The vast majority of samples (8/10) had lost Pten expression (scored as "negative" or "weak"), indicating LOH in the tumors.

We next examined if the hypomorphic *Blm*^m3/m3 context is associated with increased LOH rates at CIS genes. To this end, we

**Fig. 4** Genetic analysis of DLBCLs. **a** Scheme of experimental and analytic workflow. **b** Overlay of copy number profiles showing cancer-relevant amplifications/deletions in 16 DLBCLs from *ITP2-M;Rosa26*$^{PB/+}$;*Blm*$^{m3/m3}$ *(IPB)* mice. A zoomed-in view is provided for Chr11, which harbors frequent amplifications (identified in mouse GCB as well as ABC DLBCLs). The minimal amplified region contains—among other genes—the known DLBCL oncogenes *Bcl11a* and *Rel*. **c** Circos plot visualizing transposon insertion data from 42 IPB-DLBCLs. Rings from inward to outward: Insertion density plot for both orientations (shown in blue and green), number of insertions per common insertion site (CIS; dark yellow bars; axis from 0 to 140), number of contributing samples per CIS (red bars; axis from 0 to 40). Top 50 CIS genes are annotated. **d** Top 50 CIS genes ranked by number of contributing DLBCL samples. Molecular function (determined by literature search) of CIS genes is indicated. Transcription factors constituted the largest functional gene class (*n* = 15). Compared to the total number of mouse transcription factors (*n* = 1603), the enrichment for transcription factors among the top 50 CIS genes is highly significant (*p* = 1.4 10$^{-5}$, Fisher's exact test). Genes present in Cancer Gene Census (CGC) database and/or known for their role in DLBCL (literature search) are represented in dark blue, unknown DLBCL genes by light blue boxes. QiSeq quantitative transposon insertion site sequencing, aCGH array comparative genomic hybridization

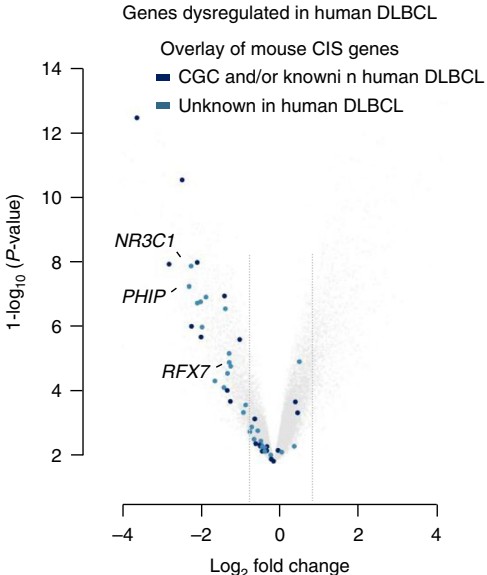

**Fig. 5** CIS genes are significantly downregulated in human DLBCL. Volcano plot shows negatively (left) and positively (right) regulated genes in human DLBCL samples relative to non-malignant B cells (centroblasts) (GSE12453). Gray lines indicate log$_2$ fold changes of −0.8 and 0.8. Dark blue colored points represent human orthologues of genes from the top 50 CIS list that are included in the Cancer Gene Census (CGC) database and/ or have already implicated roles in DLBCL. Light blue colored points depict candidate genes with unknown function in DLBCL

took advantage of a similar whole-body screen, which we performed in *Blm*-proficient mice. A critical consideration is that many tumor suppressors can act in a haploinsufficient manner, which might be cell/entity-specific. For example, it is known that *Pten* can act as a haploinsufficient TSG, but is also frequently inactivated homozygously in various cancer types. Little is known how the cellular/entity-context affects either scenario. We therefore considered it essential to look at the same entity when performing comparative LOH analyses in *Blm*$^{m3/m3}$- and *Blm*-proficient screens. In *Blm*-proficient mice BCLs are rare, but the large size of the screen (*n* = 256 tumors) allowed us to collect a sufficient number of DLBCLs (*n* = 7) for such analyses. All seven DLBCLs had high-coverage *Pten* insertions, making them suitable for side-by-side comparisons with corresponding *Pten*-altered lymphomas in the *Blm*$^{m3/m3}$ cohort.

We performed IHC-based semi-quantitative assessment of Pten expression and found that the majority of *Blm*-proficient DLBCLs (5/7) expressed substantial Pten levels, ruling out the possibility of homozygous *Pten* inactivation and LOH in these samples (Fig. 6c). In contrast, among the above described *Blm*$^{m3/m3}$ DLBCLs,

only 2/10 showed significant Pten expression, suggesting that LOH was much more common (*p* = 0.03, Fisher's exact test). Altogether, these data support a model in which the *Blm*$^{m3/m3}$ context elevates LOH at CIS genes, thereby facilitating recessive screening.

**CRISPR/Cas9-based in vivo validation of candidate genes**. To validate cancer genes emerging from our screen, we first performed in vitro competition assays using DLBCL cell lines, but did not observe significant effects upon candidate gene knockdown (Supplementary Figure 14). Functional gene validation in full-blown cancer cell lines can however be substantially limited by various factors, including (i) the wrong genetic, mutational, or cell-of origin context in which a gene of interest is operative, (ii) the highly abnormal genome and aggressive nature of full-blown cancer cells, often precluding the detection of subtle phenotypes, (iii) the restricted set of readouts in vitro, which do not encompass the large spectrum of possible cancer-driving processes affected by a gene, or (iv) inherent differences between in vitro and in vivo situations. To overcome these and other limitations related to in vitro validation experiments in cancer cell lines, we developed an efficient in vivo approach for BCL functional genomics. To this end we combined *Eμ-myc* mice (a well-established model system for B-cell lymphomagenesis[54]) with a Cas9 knock-in mouse line (*Rosa26*$^{Cas9}$ mice; Fig. 7a), which we generated in *C57BL/6* JM8 ES cells (Supplementary Figure 15) to support syngeneic transplantation experiments. We isolated E13.5 fetal liver cells from *Eμ-myc;Rosa26*$^{Cas9}$ mice as a source of hematopoietic stem and progenitor cells (HSPC), and manipulated them ex vivo using CRISPR/Cas9. HSPCs were put into short-term culture and infected with a GFP-tagged lentiviral vector containing a single guide RNA (sgRNA) cassette. Cas9 expression from the *Rosa26* knock-in allele substituted for viral Cas9 delivery. The small cargo size of sgRNA-only viral vectors allowed us to produce high viral titers and achieve high transduction rates (typically around 20%; Supplementary Figure 16). This was the key step for accomplishing an efficient validation platform, as we had only been able to achieve transduction rates of ~1% when using all-in-one (Cas9 plus sgRNA) lentiviral vectors. HSPCs were then used for reconstitution of irradiated *C57BL/6* recipient mice. We monitored transplanted animals until appearance of signs of sickness and/or tumor development.

To validate our model, we first performed side by side targeting of *Trp53* by CRISPR/Cas9 and RNAi in *Eμ-myc;Rosa26*$^{Cas9}$ HSPCs (Fig. 7b). We observed significantly accelerated lymphomagenesis in recipients transplanted with Trp53-sgRNA HSPCs or Trp53-shRNA HSPCs as compared to mice receiving HSPCs transduced with non-targeting sgRNA (control mice). Furthermore, lymphoma development occurred significantly earlier in the Trp53-sgRNA group (CRISPR/Cas9 knockout) as compared to the Trp53-shRNA (knockdown) group, reflecting gene-dosage effects.

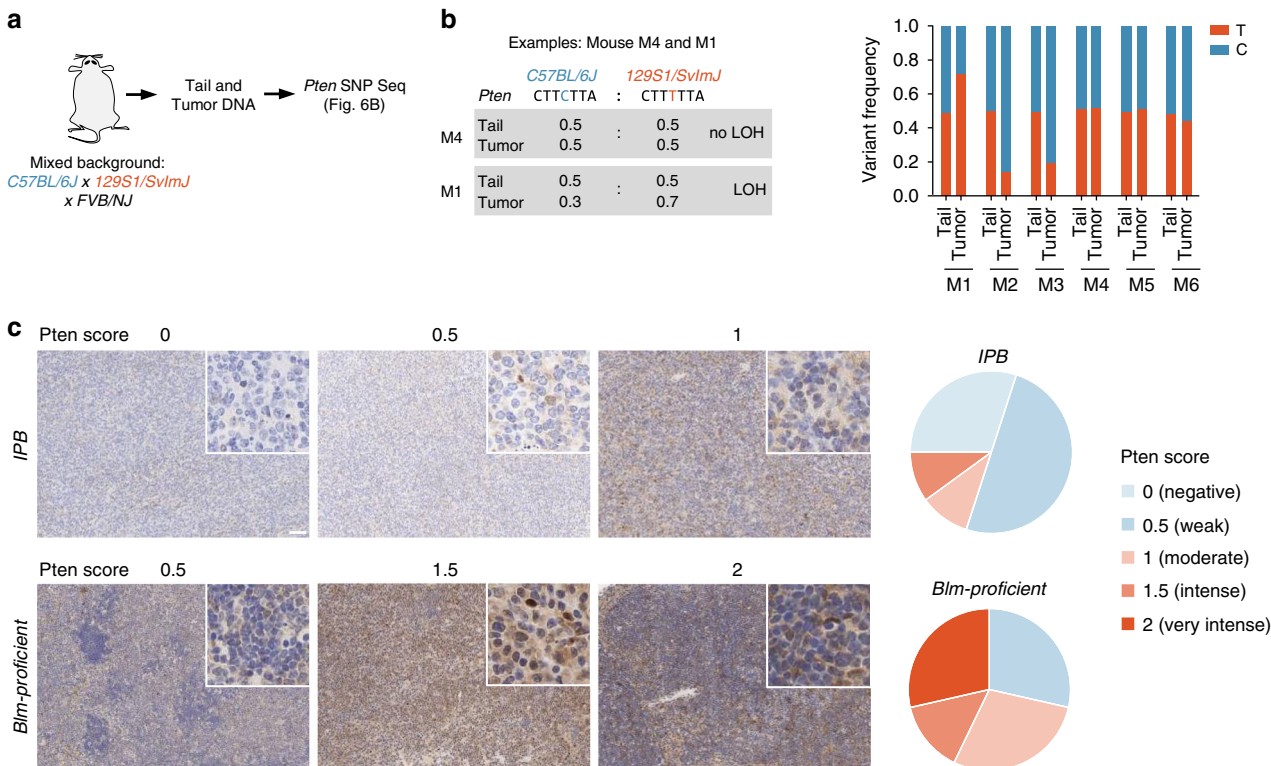

**Fig. 6** Loss of heterozygosity analysis in tumors from *IPB* mice. **a** Loss of heterozygosity (LOH) analysis workflow. Fragment of *Pten* containing the single nucleotide polymorphism (SNP) *rs30424206* was amplified and sequenced in tumors and tails from *ITP2-M;Rosa26^PB/+^;Blm^m3/m3^* (*IPB*) mice (*n* = 10). **b** Variant frequencies determined by *Pten* SNP sequencing of tail and tumor tissue samples from six *IPB* mice. All tumors harbored high-coverage *Pten* insertions. **c** Pten IHC of tumors from *IPB* mice (*n* = 10), as well as from DLBCLs generated in a *Blm*-proficient screen using *ATP2-S1;Rosa26^PB/+^* mice (*n* = 7). Pten expression was scored semi-quantitatively. Representative microscopic images are shown. Pie charts display proportions of tumor samples with different *Pten* expression scores. Scale bar, 50 μm; Insets, ×200 magnification

The strikingly high number (22/50; 44%) of genes involved in transcriptional regulation and chromatin organization identified in our screen prompted us to validate two candidate tumor suppressor genes linked to these functional groups: *Regulatory factor X7* (*Rfx7*) and *Pleckstrin homology domain interacting protein* (*Phip*). Both genes are substantially downregulated in human DLBCL as compared to non-malignant B cells (Fig. 5).

*RFX7* is a member of the *RFX* transcription factor family with unknown function. Transposon insertion patterns in *Rfx7* suggest a tumor suppressive function (Fig. 7c). In line with this observation, rare truncating mutations in *RXF7* have been reported in human BCL[9,55] (Supplementary Figure 17). CRISPR/Cas9 targeting of *Rfx7* in HSPCs confirmed its predicted function as a DLBCL tumor suppressor (Fig. 7d; Supplementary Figure 18a). Mice transplanted with Rfx7-sgRNA transduced *Eμ-myc;Rosa26^Cas9^* HSPCs showed significantly accelerated lymphomagenesis. In particular, whereas all (5/5) animals in the Rfx7-sgRNA group developed BCLs between 64 and 109 days post transplantation (pt), only one (1/9) control mouse was diagnosed with BCL at 129 days pt (Fig. 7d). Sequencing of the *Rfx7* target region in Rfx7-sgRNA tumors showed clonal frameshift-inducing insertions/deletions (indels) ranging from 1 bp to 5 bp (Supplementary Figure 19a) and real-time quantitative PCR (qPCR) revealed a significant reduction of Rfx7 expression (Supplementary Figure 18b).

PHIP is a bromodomain-containing protein expressed in a wide range of tissues, including B cells. Its molecular function has not been studied so far, but an association with melanoma progression and oncogenic effects has been reported recently[56]. The transposon insertion pattern in our mouse cancers predicts

however that gene-inactivation is the cancer-causing mechanism (Fig. 7e), suggesting a role of *Phip* as a tumor suppressor in DLBCL. To examine whether *PHIP* is relevant to human BCL, we analyzed publicly available genomic datasets derived from human BCL cases. Single nucleotide variants were rare across different studies, although occasional truncating mutations were observed[9,55,57] (Supplementary Figure 17). For one dataset copy number data derived from SNP arrays is available (TCGA-DLBC; TCGA Research Network: http://cancergenome.nih.gov). Figure 7f shows the type and frequency of copy number variation at human chromosome 6 (the home of *PHIP*) in DLBCL. Focal or arm-level Chr6q loss is frequent in human DLBCL, with several minimal commonly deleted regions, of which one affects *PHIP*. We therefore sought to functionally validate *Phip* using the CRISPR/Cas9-based approach described above. We found that mice receiving *Eμ-myc;Rosa26^Cas9^* HSPCs transduced with a Phip sgRNA (Supplementary Figure 16) presented significantly accelerated BCL development (Supplementary Figure 18a), with an onset ranging from 47 to 130 days pt (Fig. 7g and Supplementary Figure S19b). Moreover, BCLs harvested from these animals showed significantly reduced Phip expression (Supplementary Figure 18b). We thus conclude that *PHIP* is a target gene in the minimal deleted region on chromosome 6 in human DLBCL. Altogether, these data validate *Rfx7* and *Phip* as potent tumor suppressors in B-cell lymphomagenesis.

**Clinical relevance of the identified tumor suppressor genes.** To investigate if TSGs from the top 50 CIS list can be used as predictive markers in human DLBCL, we systematically analyzed their expression in a large clinically annotated DLBCL patient

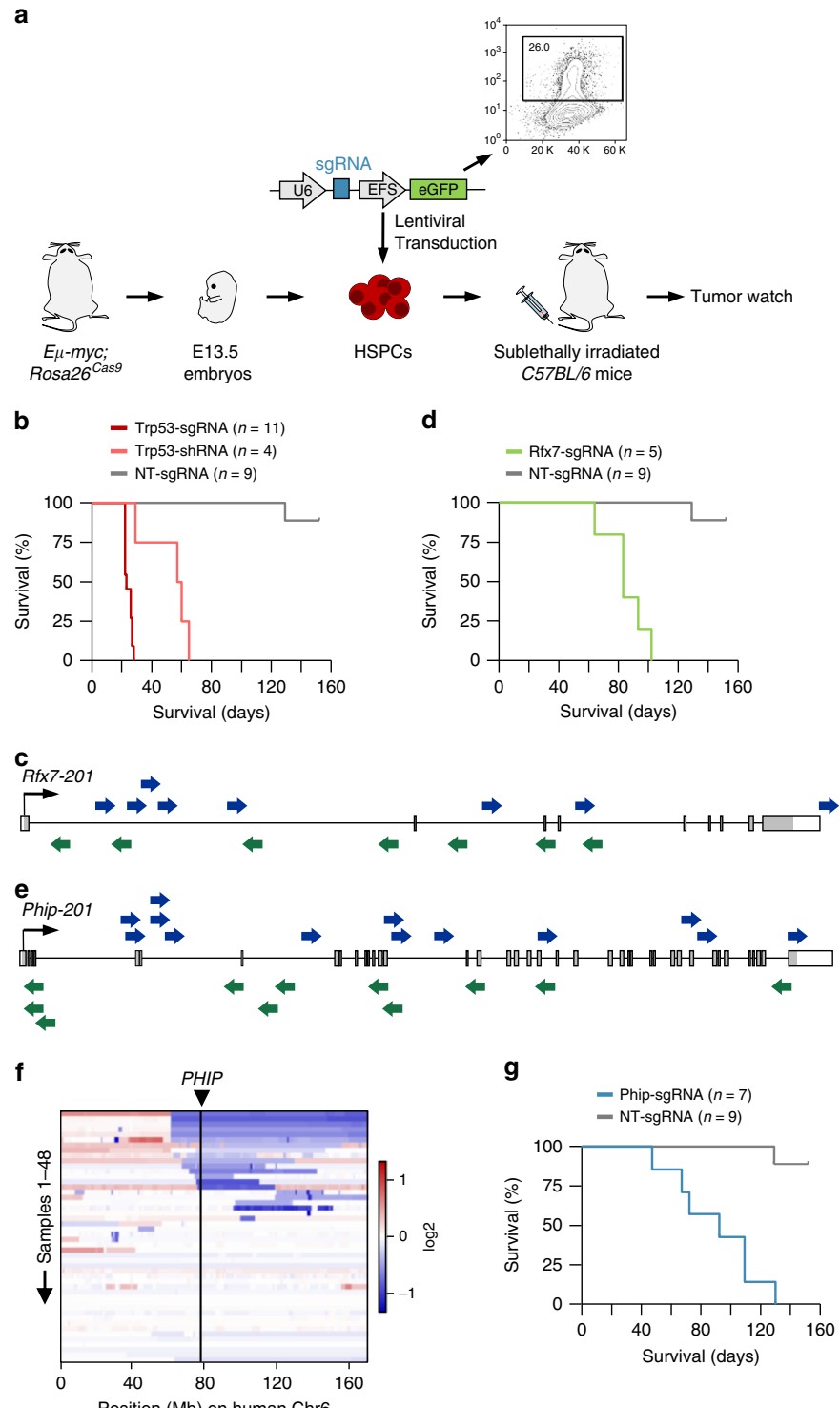

cohort ($n = 424$; GSE31312). Patients were diagnosed with de novo DLBCL according to WHO criteria and were monitored as part of the "International DLBCL Rituximab-CHOP Consortium Program Study" in 29 medical centers[58]. Gene expression profiles were generated prior to standard treatment with R-CHOP (rituximab, cyclophosphamide, doxorubicin, vincristine, prednisone). Notably, we observed a significant over-representation of predictors of poor survival in the top 50 CIS list ($p = 2.97 \times 10^{-5}$, Fisher's exact test): for 11 genes, patients with cancers expressing low mRNA levels had a significantly reduced overall survival (Supplementary

Table 4). Effects on survival could also be observed in another, albeit smaller, DLBCL cohort ($n = 220$; GSE10846) for 10 out of these 11 genes. Kaplan–Meier plots displaying overall and progression-free survival for the top five genes are shown in Fig. 8. Among them is *NR3C1*, which codes for the glucocorticoid receptor, a druggable gene (for information on the druggability of other CIS genes see Supplementary Data 7). The glucocorticoid prednisone is part of the standard treatment regime of different hematopoietic cancer types. In T-cell acute lymphoblastic leukemia resistance to glucocorticoid treatment has been previously

**Fig. 7** A CRISPR/Cas9-based in vivo platform for BCL functional genomics. **a** Outline of the functional genomic approach for in vivo gene validation. **b** Kaplan–Meier plot showing survival of mice transplanted with hematopoietic stem/progenitor cells (HSPC) transduced with *Trp53* sgRNA, *Trp53* shRNA, and non-targeting (NT) sgRNA. *p* < 0.0001 for both Trp53-sgRNA vs. NT-sgRNA and Trp53-shRNA vs. NT-sgRNA. *p* < 0.03 for Trp53-sgRNA vs. Trp53-shRNA. *p*-values (log-rank test) were corrected for multiple testing using the Bonferroni method. **c** The transposon insertion pattern in *Rfx7* predicts that gene-disruption is the cancer-causing mechanism. Each arrow represents an individual insertion. Insertions from all DLBCLs in the cohort are shown. *ITP2* transposons can trap genes in either orientation. Insertions are distributed over the whole length of the gene (predicting a tumor suppressor). There is no bias for hot-spot areas of insertions as typically observed for unidirectional "activating" insertions in oncogenes. Consensus coding sequence (*Rfx7-201*) is displayed. **d** Kaplan–Meier plot for mice transplanted with HSPCs transduced with Rfx7 sgRNA and NT sgRNA. *p* < 0.0001, log-rank test. **e** Transposon insertion pattern in *Phip*, predicting tumor suppressive function of the gene. Consensus coding sequence (*Phip-201*) is shown. Each arrow represents an individual insertion. Insertions from all DLBCLs in the cohort are shown. **f** Heatmap displaying copy number variations on human chromosome 6 in samples from the TCGA-DLBC (*n* = 48) dataset (TCGA Research Network: http://cancergenome.nih.gov). The position of *PHIP* is indicated. **g** Function of *Phip* as a B-cell lymphoma tumor suppressor was validated using the CRISPR/Cas9-based in vivo functional genomic approach. Kaplan–Meier plot for mice transplanted with HSPCs transduced with Phip sgRNA and non-targeting (NT) sgRNA transplants. *p* < 0.0001, log-rank test. sgRNA single guide RNA, EFS elongation factor 1-alpha core promoter, eGFP enhanced green fluorescent protein, Mb megabase

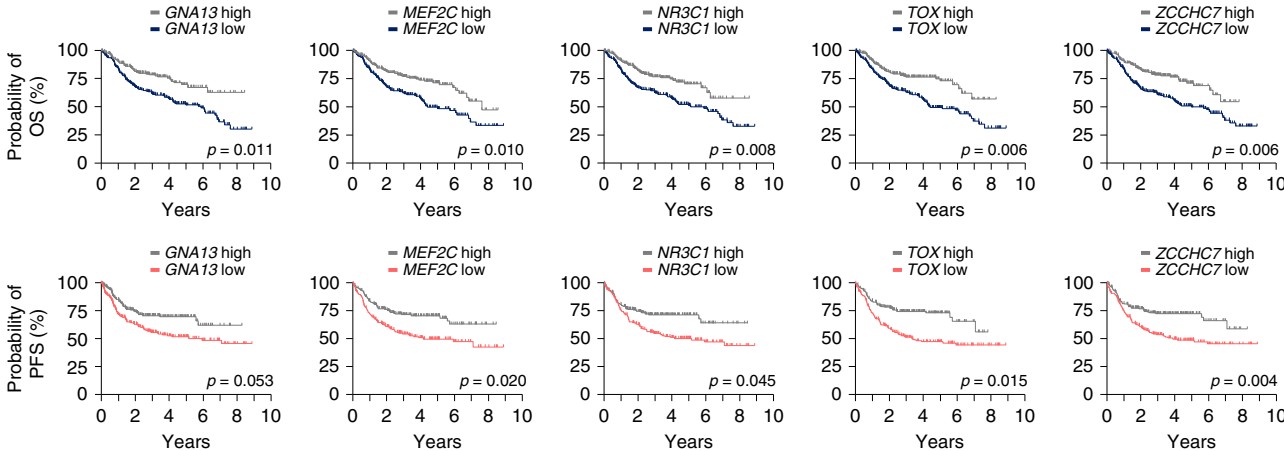

**Fig. 8** Clinical relevance of CIS genes in human DLBCL. Association of CIS genes with overall and progression-free survival in a large clinically annotated DLBCL patient cohort (*n* = 424; GSE31312). For each of the 50 CIS genes, the cohort was stratified into "low" (LE; below median expression) or "high" expression (HE, above median). Kaplan–Meier plots for the top five genes, for which associations (between low gene expression and poor survival) were also observed in a second DLBCL patient cohort (*n* = 220; GSE10846), are shown. The significance threshold was set to a false discovery rate of 0.05. For genes, for which multiple probes were available, the one giving the signal with the highest variance was selected. Association with overall survival (OS) is shown on the top, and association with progression-free survival (PFS) on the bottom

associated with SNPs or deletions in *NR3C1*[59]. Although genetic alterations in *NR3C1* are very rare in human DLBCLs, its expression levels vary significantly between patients. Finally, our screen also discovered genes that are interacting with or downstream of *Nr3c1*, including the CIS genes *Crebbp*, *Pou2f1*, and *Hnrnpu*, further supporting the relevance of this pathway in DLBCL biology. Taken together, our discovery of *Nr3c1* as a major DLBCL CIS and its strong association with therapy response/resistance establishes *NR3C1* as a bona fide tumor suppressor and predictive marker in human DLBCL.

## Discussion

Our work describes transposon and CRISPR tools/methods for gene discovery as well as functional annotation of cancer genomes. Their application provided comprehensive insights into the molecular landscapes of DLBCL. We performed the first BCL-focused *PB* transposon screen and discovered large sets of DLBCL drivers. More than half of these genes have not been found earlier by other approaches to cancer genome analysis. This study thus vastly expands the catalog of human DLBCL genes. Multiple layers of evidence support the validity and importance of the screening results, including (i) the enrichment of bona fide

DLBCL tumor suppressors among the top-scoring genes in our list (e.g., *Gna13*), (ii) the positive validation of discovered genes, such as *Rfx7* and *Phip*, which we show to indeed suppress lymphomagenesis in vivo, (iii) the integration of human data, which showed that tumor suppressors identified in our screen are indeed dysregulated in human DLBCL, and (iv) the discovery of cancer genes with clinical relevance, including biomarkers predicting treatment response and survival in human DLBCL. All together, these data underline the power of our screening approach to inform BCL biology.

## Methods

**Generation of mouse strains and cohorts.** Constitutive *PB* transposase knock-in mice (*Rosa26^PB*), *Blm^m3* mice, and *Eμ-myc* transgenic mice have been described earlier[20,33,54].

To generate *Rosa26^Cas9* knock-in mice (*Gt(ROSA)26Sor^tm1(Cas9)Rrad*), we cloned a human codon-optimized hemagglutinin-tagged Cas9 sequence derived from *Streptococcus pyogenes* into a Gateway-compatible entry vector (Addgene #17398[60]), which was then shuttled into a *Rosa26*-targeting Gateway destination vector with loxP-flanked puromycin resistance-containing stop cassette (modified after Addgene #21189[60]). Embryonic stem cell (JM8) targeting, blastocyst injections, and subsequent breeding steps were performed using standard protocols/techniques. Upon generation of conditional *Rosa26^LSL-Cas9* knock-in mice, we derived a constitutive *Rosa26^Cas9* mouse line by deletion of the loxP-flanked stop cassette in the germline. *Rosa26^LSL-Cas9* and *Rosa26^Cas9* mice were

established and maintained on a pure *C57BL/6* background. Genotyping primers for *Rosa26^Cas9* knock-in mice are listed in Supplementary Data 8.

Transgenic transposon mouse lines harboring *ITP1* and *ITP2* transposons were generated as delineated in Rad et al.[20]. Briefly, ITP transposons were cut out of the *pBlueScript* donor vector with appropriate restriction enzymes and prepared for pronuclear injection using standard techniques. Founder animals were screened for transposon integrations by Southern blot using an *En2SA*-specific probe. Metaphases derived from the peripheral blood of the animals were prepared to identify transposon donor loci by fluorescence in situ hybridization (FISH) and quantitative real-time PCR was conducted to determine the transposon copy number in the founder mice with primer and probe sequences listed in Supplementary Data 8. Founder animals were then crossed with *C57BL/6* mice and offspring was genotyped by PCR using previously described protocols[28] with primers listed in Supplementary Data 8. F2 animals were used to establish transgenic transposon mouse lines and FISH analysis confirmed transposon integration sites in these animals after death on metaphases from spleen preparations.

Experimental *(ITP2-M;Rosa26^PB/+;Blm^m3/m3)* and control *(Rosa26^PB/+; Blm^m3/m3* and *ITP2-M;Blm^m3/m3)* mice were maintained on a mixed C57BL/6J x 129S1/SVImJ x FVB/NJ background.

Note that triple-transgenic *ITP1-C;Rosa26^PB/+;Blm^m3/m3* mice showed extensive embryonic lethality, which is primary linked to the large size of the *ITP1* transposon concatemer (*n* = 70 copies) for following reasons: (i) *Blm^m3/m3* mice are fertile and viable and develop no embryonic phenotypes[33], excluding the *Blm^m3/m3* allele as cause of the observed lethality, (ii) double transgenic *ATP; Rosa26^PB/+* mice with transposon copy numbers similar to *ITP1-C;Rosa26^PB/+* mice showed comparable or even higher rates of embryonic lethality in a Bloom-proficient background[20], and (iii) in *ITP2-M;Rosa26^PB/+;Blm^m3/m3* mice, which harbor half as much transposon copies, embryonic lethality is dramatically reduced.

Mice were kept in the animal facilities of the Wellcome Trust Sanger Institute, Hinxton/Cambridge, UK and Klinikum rechts der Isar, Technical University Munich, München, Germany under specific-pathogen-free (SPF) conditions on a 12-h light/dark cycle, receiving food and water ad libitum. All animal experiments were carried out in compliance with the requirements of the European guidelines for the care and use of laboratory animals and were approved by the UK Home Office and the Institutional Animal Care and Use Committees (IACUC) of the Technical University Munich (Regierung von Oberbayern, Munich, Germany).

**Tests of splice acceptor elements**. Gene trapping efficiency tests were performed as described in Prosser et al.[61]. Briefly, transposons carrying different splice acceptor elements were cloned into a hypoxanthine phosphoribosyltransferase (Hprt) exchange vector between a loxP and a lox511 cassette in both orientations. These Hprt exchange vectors were electroporated individually together with a Cre expression vector into male embryonic stem cells, which harbor an acceptor cassette knock-in at the X-chromosomal Hprt locus. Recombinase-mediated cassette exchange led to replacement of a resistance marker cassette (CMV-EM7-BSD-pA) with the transposon-containing exchange cassette (elements between the loxP and lox511 cassette in the Hprt vector). Embryonic stem cells were then selected with 6TG. Efficient splicing by the splice acceptor within the transposon leads to a premature termination of transcription resulting in a non-functional Hprt protein. Hprt deficiency induces 6TG resistance. After selection, cells were stained with crystal violet and 6TG-resistant colonies were counted.

**Necropsy and histopathological analysis**. All animals were monitored regularly for signs of sickness (e.g., inactivity, palpable/visible masses, and poor grooming). During necropsy, a gross inspection of all internal organs was carried out. For DNA/RNA isolation, tissue samples were stored in RNAlater (Sigma). For histology, tissue samples were fixed in 4% formaldehyde overnight, paraffin-embedded, sectioned, and stained using hematoxylin and eosin following standard protocols. Experienced mouse hematopathologists, who were blinded to the mouse genotypes, performed analysis of the tumors.

**DNA and RNA isolation**. DNA and RNA isolation of tissue samples was performed according to manufacturer's instructions using the Qiagen DNeasy Blood & Tissue Kit and the Qiagen RNeasy Plus Mini Kit, respectively.

**cDNA synthesis and real-time qPCR**. cDNA synthesis was conducted using SuperScript II Reverse Transcriptase (Thermo Fisher Scientific) according to standard protocols. Real-time qPCR was conducted with SYBR™ Select Master Mix (Thermo Fisher Scientific) and primers listed in Supplementary Data 9.

**Immunohistochemistry**. IHC was performed with antibodies directed against B220/CD45R (RA3-6B2; R&D Systems; 1:40 dilution), CD138 (281-2; BD Biosciences; 1:50 dilution), CD3 (A0452; DAKO; 1:100 dilution), myeloperoxidase (A0398; DAKO; 1:100 dilution), Ki-67 (RM-9106-S1; Thermo Fisher Scientific; 1:200), Bcl6 (sc-858; Santa Cruz Biotechnology; 1:50), Irf4 (sc-6059; Santa Cruz Biotechnology; 1:100), and Pten (M362729-2; Agilent; 1:150 dilution).

Pre-treatment of sections was conducted with EDTA for 20 min (B220, CD138, CD3, myeloperoxidase), 30 min (Ki-67, Irf4, Pten), or 40 min (Bcl6).

We used goat anti-rabbit, rabbit anti-rat, rabbit anti-goat, and rabbit anti-mouse secondary antibodies (AffiniPure Goat Anti-Rabbit IgG (H+L) [111-005-003; Jackson ImmunoResearch], AffiniPure Rabbit Anti-Rat IgG [312-005-045; Jackson ImmunoResearch], Rabbit Anti-Goat IgG [P0449; DAKO], and Rabbit Anti-Mouse IgG [ab125904; Abcam]).

**RNA sequencing**. Library preparation for bulk 3′-sequencing of poly(A)-RNA was done as described in Parekh et al.[62]. Briefly, barcoded cDNA of each sample was generated with a Maxima RT polymerase (Thermo Fisher Scientific) using oligo-dT primer containing barcodes, unique molecular identifiers (UMI) and an adapter. 5′ ends of the cDNAs were extended by a template switch oligonucleotide (TSO). After pooling of all samples, full-length cDNA was amplified with primers binding to the TSO-site and the adapter. cDNA was tagmented with the Nextera XT kit (Illumina) and 3′-end-fragments finally amplified using primers with Illumina P5 and P7 overhangs. In comparison to Parekh et al. (2016), the P5 and P7 sites were exchanged to allow sequencing for the cDNA in read1 and barcodes and UMIs in read2 to achieve a better cluster recognition. The library was sequenced on a NextSeq 500 (Illumina) with 75 cycles for the cDNA in read1 and 16 cycles for the barcodes and UMIs in read2.

The minor murine reference genome release *GRCm38.p6* including all haplotypes and patches was used as reference for mapping the raw read data with *Dropseq tools v1.13*. Gencode annotation release *M19* was used to determine read counts per gene. The resulting *genes x samples* count matrix was imported into *R v3.4.4* and further processed with *DESeq2 v1.8*. Prior clustering, lowly expressed genes were removed and the data were subsequently rlog transformed with a parametric fit and an intercept only design. Gene lists used for classification of samples were taken from Wright et al.[40]. Genes with murine orthologues were used for clustering of rlog transformed expression data, using the Ward method for cluster agglomeration. Z-transformed expression values are shown as heatmap.

**B-cell receptor repertoire sequencing**. Analysis of the immune repertoire of murine diffuse large B-cell lymphoma samples was performed as described in Turchaninova et al.[63]. with minor modifications, which allowed us to introduce Illumina Nextera adapters and indexes during PCR[64]. Briefly, 700 ng RNA (extracted from whole tissue lysate) was transcribed into cDNA using a 5′ template switch oligo (TSO), which contains a unique molecular identifier. IGH and IGL libraries were amplified using a set of IGHC-specific/IGLC-specific and 5′ TSO-specific primers introducing indexed Nextera sequencing adapters (listed in Supplementary Data 10). The resulting libraries were analyzed on the Illumina MiSeq (300 bp paired end).

**Analysis of the B-cell receptor repertoire**. For analysis of the B-cell receptor repertoire, raw data were demultiplexed using *bcl2fastq*. Unique molecular identifier (UMI) tag extraction was performed using the *CheckoutBatch* command from *MIGEC 1.2.9*[42] with options *-cute* and *--overlap* set and remaining parameters left as default values. UMI-guided consensus assembly was conducted according to the default settings of *MIGEC* in accordance with MIG statistics derived by the same tool. This led to different MIG size thresholds used for different samples, with some samples having a low threshold. For further analysis, only overlapping read pairs were considered. IGH chain and IGL chain samples were analyzed separately using *MiXCR 2.1.12* software[43]. For each sample, reads were aligned to the mouse reference sequences provided by *MiXCR*. The chain type for the heavy chain samples was specified as "IGH" and for the light chain as "IGL,IGK". To allow downstream processing following non-default parameters were used: *--save-description*, *--save-reads*, *-p rna-seq*, *-OreadsLayout=Unknown*. Clone assembly was based on the CDR3 region. An index file was generated in the process; *-OsearchDepth* parameter was set to 0. Information about the clones was exported using the *exportClones* command and following non-default parameters: *--preset full*, *-cGenes*, *-o*, *-t* (therefore excluding out-of-frame and stop codon containing mutants) and *-chains* specified according to the sample type. Furthermore alignment information was exported using the *exportAlignment* command with parameters *-descrR1*, *-readID*, *-vBestIdentityPercent* and *-cloneID* set. For all subsequent analyses, we further excluded clusters that were supported by a single unique UMI-labeled cDNA molecule sequence. This was performed to increase the confidence of the final clonality analysis, as most of the sequence subvariants which might originate from errors obtained in the course of cDNA synthesis or from under-corrected PCR and sequencing errors remaining after UMI consensus assembly were removed[63].

**Generation of clonality network plots**. For visualization of the clonal structures of individual tumos, we developed *CloNet*, a pipeline for generation of clonality network plots. Each dot in the plot represents a clone reported by *MiXCR*. The size of the dot scales with the third root of the number of reads assigned to the respective clone. Note that due to such scale the differences at high read counts appear less pronounced than in the small and middle range. This definition was necessary to enable visualization of the dominant clones in monoclonal samples, which can accumulate more than 90% of all reads. Two clones were connected by a

link if they map to at least one equal V and J gene and differ by only 1 bp in their CDR3 sequence. A fully connected group of clones forms a clonal cluster. A clonal cluster was highlighted by color if the cluster accumulated 10% or more of the overall read counts of the sample.

**Array comparative genomic hybridization**. Array comparative genomic hybridization (aCGH) was performed as described in Wolf et al.[65]. Briefly, Agilent oligonucleotide aCGH (Agilent 60 k mouse CGH arrays with a custom design (AMADID 041078)) was conducted according to manufacturer's instructions. Agilent Genomic Workbench software version 7.0.4.0 was used for pre-processing of aCGH data. For aberration calling, the ADM-2 algorithm was applied. Segment coordinates were reported for the GRCm37 reference genome.

**Quantitative transposon insertion site sequencing**. QiSeq was performed as described in Friedrich et al.[28]. Briefly, DNA samples were sheared with a Covaris AFA sonicator to a mean fragment length of 250 bp. The fragmented DNA was then end-repaired, A-tailed, and a splinkerette adapter was ligated to each DNA end. For the 5′ and 3′ transposon end, subsequent steps (amplification and sequencing of transposon–genome junctions) were conducted separately. The specific structure of the splinkerette adapter (Y-shaped design with a template and a hairpin strand) ensures that only transposon–genome junction fragments (and not genomic fragments without transposon insert) can be amplified in the following first PCR step (which was conducted with transposon-specific and splinkerette-specific primers). Afterwards, a second nested PCR step was performed for further amplification, barcoding of samples and extension with Illumina flow cell-binding sites P5 and P7. Each sample was then quantified with quantitative real-time PCR (using P5-specific and P7-specific primers). Subsequently, samples were equimolarly mixed and the library pool was again quantified. Libraries were sequenced on the Illumina MiSeq sequencer (75 bp, paired-end). Mapping of integrations to the mouse genome was performed using the *SSAHA2* algorithm and sequences containing transposon–genome junctions were selected for downstream analyses.

**CIMPL analysis**. For identification of CIS (genomic regions that were more frequently hit by transposons than expected by chance), *ITP2* insertions with a read coverage ≥20 were subjected to statistical analysis using CIMPL (Common Insertion site Mapping PLatform) analysis, which is based on a Gaussian kernel convolution framework[50]. Insertions within a 3 Mb region upstream and downstream of the transposon donor locus were excluded from the analysis (local hopping area of the transposon as described in Rad et al.[20]). We used different scale parameters (30,000, 50,000, 70,000, and 90,000) and only included CISs identified across all scales and being supported by at least 10% of the samples in the analysis. CISs were ranked according to the number of contributing tumor samples. *Sfi1*, a known artifact frequently detected in insertional mutagenesis screens[66], was removed from the list of CIS genes.

**TAPDANCE analysis**. For TAPDANCE (Transposon Annotation Poisson Distribution Association Network Connectivity Environment) analysis[51], *ITP2* insertions with a read coverage ≥2 were considered. As for CIMPL analysis, insertions 3 Mb region upstream and downstream of the donor locus on chromosome 14 were excluded and *Sfi1* was removed from the CIS gene list. All top 50 CIS genes identified by CIMPL were also detected with TAPDANCE analysis.

**Co-occurrence analysis**. A Fisher's exact test was performed for co-occurrence interference of the top 50 CIS genes. *p*-values were corrected for multiple testing (Benjamini–Hochberg).

**Cross-species analyses**. Analyses were performed using the publicly available datasets Reddy et al.[4] and the TCGA DLBCL dataset (TCGA Research Network: http://cancergenome.nih.gov).

**Gene expression analyses**. For gene expression analyses, datasets with the accession numbers GSE12453 (only DLBCL and normal B cells (centroblasts) were used for analysis), GSE12195 and GSE2350 were retrieved from the NCBI Gene Expression Omnibus database. Raw data was normalized with *RMA*. Gene annotations for each probe set were derived from the *Ensembl* v90 database. If multiple probe sets represent the same gene, the probe set with the highest mean intensity across all samples for a given dataset was used for further analyses. Differential expression between conditions of interest were tested with *limma*. A gene was considered to be significant if the false discovery rate was <0.05 and the log$_2$ expression change between conditions was at least 0.8.

**Sequencing of regions with SNPs**. A SNP (*rs30424206*)-containing region within the mouse *Pten* gene was amplified and sequenced as described in Weber et al.[67]. Briefly, genomic CRISPR/Cas9 target regions were amplified with Q5® High-Fidelity DNA Polymerase (New England Biolabs) using PCR primers listed in Supplementary Data 9. PCR products were purified with the Monarch® PCR &

DNA Cleanup Kit (5 μg) (New England Biolabs). For library preparation, end repair and A-tailing was performed (NEBNext® Ultra™ II DNA Library Prep Kit for Illumina®, New England Biolabs) and an Illumina paired end adapter was ligated. Individual samples were barcoded with eight cycles of PCR. Barcoded samples were pooled and quantified with qPCR. The single pool was sequenced on the Illumina MiSeq (300 bp, paired end). Raw reads were preprocessed with *Trimmomatic v0.36* with the following parameter settings: leading/trailing Phred quality score cut-off: 25; minimum read length: 50 nt; minimum average Phred quality score within a sliding window of 10 nt: 25. Forward and reverse reads passing these filters were combined using *Flash v.1.2.11*. Merged reads were mapped to the *mm10* mouse reference genome following variant calling with *BBMap* (https://sourceforge.net/projects/bbmap). Only variant positions with a coverage of at least 100 reads were considered for downstream analyses.

**Cell-culture-based competition assays**. For in vitro knockdown experiments of candidate genes, *RFX7* and *PHIP* shRNAs and a scrambled control shRNA (sequences listed in Supplementary Data 9) were cloned into a lentiviral pLKO.1 vector containing a U6 promoter-driven shRNA cassette and a blue fluorescent protein (BFP) driven by the phosphoglycerate kinase promoter. The GCB-DLBCL cell line HT (ATCC® CRL-2260™) and the ABC-DLBCL cell line RIVA (ACC 585) were cultivated according to distributor's instructions and lentivirus production was conducted according to standard protocols. For each shRNA construct, $5 \times 10^5$ cells were transduced with the respective lentiviral particles in a well of a 12-well-plate using a spin infection protocol. For competition assays, transduced BFP-positive cells were co-cultured with non-transduced cells in six-well-plates and the proportion of BFP-positive cells was analyzed on day 3, 7, 10, 14, and 17 post infection. Percentage of BFP-positive cells was normalized to day 3 post transduction. For analysis of knockdown efficiencies, BFP-positive HT and RIVA cells were sorted 17 days post infection. *RFX7* and *PHIP* expression was determined by real-time quantitative PCR (qPCR) using primers specific for *RFX7* and *PHIP* transcripts (sequences listed in Supplementary Data 9). For normalization of RNA input, GAPDH qPCR (primers listed in Supplementary Data 9) was performed.

**CRISPR/Cas9-based BCL in vivo validation platform**. For the CRISPR/Cas9-based BCL in vivo validation platform, sgRNA sequences for gene targeting were selected with the Benchling CRISPR sgRNA design tool (https://benchling.com/crispr). SgRNA oligonucleotides were cloned into the *pLKO5.sgRNA.EFS.GFP* vector (Addgene #57822[68]). The plasmid contains a U6-driven sgRNA expression cassette and the fluorescence marker enhanced green fluorescent protein (eGFP) driven by the elongation factor 1-alpha core (EFS) promoter. Sequences of sgRNAs are listed in Supplementary Data 9 and on-target editing efficiencies of all sgRNAs were determined in in vitro cell culture systems before their use in vivo. Sequence and vector used for the *Trp53* short hairpin RNA (shRNA) experiments have been published previously[69].

Syngeneic transplantation experiments were conducted as described in Hoellein et al.[70]. Briefly, homozygous *Rosa26$^{Cas9/Cas9}$* and heterozygous *Eμ-myc* mice were crossed and fetal liver cells (FLC) were isolated from double-transgenic *Rosa26$^{Cas9}$*; *Eμ-myc* embryos on day E13.5. FLCs were put into short-term culture and transduced twice with *pLKO5.sgRNA.EFS.GFP* lentiviral particles in a 12-h interval. 24 h after the last infection, flow cytometry was performed to determine the percentage of eGFP-positive FLCs (usually ranging between 10% and 20%). Syngeneic C57BL/6 recipient mice were irradiated (8.5 Gy) and $2.5 \times 10^5$ eGFP-positive FLCs and $2 \times 10^5$ CD45.1 bone marrow cells (as support) were injected into the lateral tail vein. Transplanted animals were monitored regularly for signs of morbidity and lymphoma development. DNA was isolated from tumors for insertion/deletion (indel) analyses and histological analyses of processed tumor samples (which identified the cancers as lymphomas of B-cell origin) was performed.

Note that as with other murine CRISPR/Cas models, the potential immunogenicity of Cas9 has to be carefully considered. Therefore, use of Cas9 expressing recipient animals or inducible Cas9 systems might be advantageous.

**Sequencing of CRISPR/Cas9 target regions**. CRISPR/Cas9 target regions were sequenced by Sanger capillary sequencing and amplicon-based NGS as described in Weber et al.[67]. Briefly, genomic CRISPR/Cas9 target regions were amplified with Q5® High-Fidelity DNA Polymerase (New England Biolabs) using PCR primers listed in Supplementary Data 9. PCR products were purified with the Monarch® PCR & DNA Cleanup Kit (5 μg) (New England Biolabs). For library preparation, end repair and A-tailing was performed (NEBNext® Ultra™ II DNA Library Prep Kit for Illumina®, New England Biolabs) and an Illumina paired end adapter was ligated. Individual samples were barcoded with eight cycles of PCR. Barcoded samples were pooled and quantified with qPCR. The single pool was sequenced on the Illumina MiSeq (300 bp, paired end). For processing of data and downstream analysis, see "Sequencing of regions with SNPs".

**Flow cytometry**. Flow cytometry data were acquired on a FACSCanto II flow cytometer (BD Biosciences) or CytoFLEX S (Beckmann Coulter) cytometer. Sorting of cells was performed using a FACSARIA III cell sorter (BD Biosciences).

FlowJo (Tree Star Inc.) and Kaluza Software (Beckmann Coulter) were used for data analysis.

**Survival analysis**. Gene expression datasets with the accession numbers GSE31312 and GSE10846 were retrieved from the NCBI Gene Expression Omnibus database. Raw data was normalized with *VSN*. Gene annotations and filtering was performed as described above. Gene expression was median stratified before being subjected to log-rank testing with the R package *survival v2.41-3*. The false discovery rate was calculated according to the Benjamini–Hochberg procedure.

**Statistical analyses**. All statistical analyses were performed using *R v3.4.4*. Methods used for statistical hypothesis testing are directly stated in the text or figure legends. In general, the significance level was set to 0.05 and, if necessary, correction for multiple testing was applied.

**Reporting summary**. Further information on experimental design is available in the Nature Research Reporting Summary linked to this article.

## Data availability
Sequence data have been deposited at EBI European Nucleotide Archive under accession numbers PRJEB31030 and PRJEB31031.

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

## Acknowledgements
We thank Julia Eichinger, Anja Seemann, Teresa Stauber, Danijela Heide, and Olga Seelbach for excellent technical assistance. J.d.l.R. was a recipient of a FEBS Long-Term Fellowship during part of this work. M.A.T. and D.M.C. are supported by grant of the Ministry of Education and Science of the Russian Federation (14.W03.31.0005), U.K. is supported by Deutsche Krebshilfe (111305; 111944) and the German Research Foundation (SFB 1335; project P3), J.C. is supported by the Fundación María Cristina Masaveu Peterson, and R.R. is supported by Deutsche Krebshilfe (70112480), the German Research Foundation (RA 1629/2-1; SFB1243; SFB 1321), and the German Cancer Consortium Joint Funding Program.

## Author contributions
G.S.V., J.C., A.B., and R.R. designed research; J.W., J.d.l.R., C.S.R., M.S., L.R., A.St., A.P., M.J.F., R.L., R.Ö., R.B., A.Sc., J.H., L.B.K., T.A., and F.Y. performed research, I.G.M., M.M., U.Ko., G.J.H., K.S., and L.Q.-M. performed pathohistological analyses; M.A.T. and D.M.C. developed mouse immunoglobulin profiling protocol; J.W., J.d.l.R., C.S.G., O.B., T.E., M.G., G.L., G.S.V., J.C., A.B., and R.R. analyzed data; J.K., G.S., K.U., U.Z.-S., M.H., M.S.-S., F.Y., D.S., P.L., and U.Ke. contributed analytic tools/reagents; J.W. and R.R. wrote the paper.

## Additional information

**Competing interests:** The authors declare no competing interests.

Julia Weber[1,2], Jorge de la Rosa 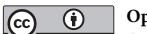[3], Carolyn S. Grove[3,4,5], Markus Schick[6], Lena Rad[3], Olga Baranov[1,2], Alexander Strong[3], Anja Pfaus[1,2], Mathias J. Friedrich[1,2,3,7], Thomas Engleitner[1,2], Robert Lersch[1,2], Rupert Öllinger[1,2], Michael Grau[8,9], Irene Gonzalez Menendez[10], Manuela Martella[10], Ursula Kohlhofer[10], Ruby Banerjee[3], Maria A. Turchaninova[11,12,13], Anna Scherger[6], Gary J. Hoffman[3,4], Julia Hess[14], Laura B. Kuhn[15], Tim Ammon[2,6], Johnny Kim[16,17], Günter Schneider[7], Kristian Unger[14], Ursula Zimber-Strobl[15], Mathias Heikenwälder[18], Marc Schmidt-Supprian[2,6], Fengtang Yang[3], Dieter Saur[2,7,19], Pentao Liu[3,20], Katja Steiger[21], Dmitriy M. Chudakov[11,12,13,22,23], Georg Lenz[8,9], Leticia Quintanilla-Martinez[10], Ulrich Keller[6,24], George S. Vassiliou[3,25,26], Juan Cadiñanos[27,28], Allan Bradley[3] & Roland Rad[1,2,7,19]

[1]Institute of Molecular Oncology and Functional Genomics, TUM School of Medicine, Technische Universität München, Munich 81675, Germany. [2]Center for Translational Cancer Research (TranslaTUM), TUM School of Medicine, Technische Universität München, Munich 81675, Germany. [3]The Wellcome Trust Sanger Institute, Genome Campus, Hinxton, Cambridge CB10 1SA, UK. [4]School of Medicine, University of Western Australia,

Crawley 6009, Australia. [5]Department of Haematology, PathWest and Sir Charles Gairdner Hospital, Queen Elizabeth II Medical Centre, Nedlands 6009, Australia. [6]Department of Medicine III, Klinikum rechts der Isar, Technische Universität München, Munich 81675, Germany. [7]Department of Medicine II, Klinikum rechts der Isar, Technische Universität München, Munich 81675, Germany. [8]Department of Medicine A, University Hospital Münster, Münster 48149, Germany. [9]Cluster of Excellence EXC 1003, Cells in Motion, Münster 48149, Germany. [10]Institute of Pathology and Comprehensive Cancer Center, Eberhard Karls Universität Tübingen, Tübingen 72076, Germany. [11]Laboratory of Genomics of Antitumor Adaptive Immunity, Privolzhsky Research Medical University, Nizhny Novgorod 603005, Russia. [12]Genomics of Adaptive Immunity Department, Shemyakin-Ovchinnikov Institute of Bioorganic Chemistry, Russian Academy of Science, Moscow 117997, Russia. [13]Pirogov Russian National Research Medical University, Moscow 117997, Russia. [14]Helmholtz Zentrum München, Research Unit Radiation Cytogenetics, Neuherberg 85764, Germany. [15]Helmholtz Zentrum München, Research Unit Gene Vectors, Munich 81377, Germany. [16]Department of Cardiac Development and Remodeling, Max-Planck-Institute for Heart and Lung Research, Bad Nauheim 61231, Germany. [17]German Center for Cardiovascular Research (DZHK), Rhine Main, Germany. [18]Divison of Chronic Inflammation and Cancer, German Cancer Research Center (DKFZ), Heidelberg 69120, Germany. [19]German Cancer Consortium (DKTK), German Cancer Research Center (DKFZ), Heidelberg 69120, Germany. [20]Li Ka Shing Faculty of Medicine, Stem Cell and Regenerative Medicine Consortium, School of Biomedical Sciences, University of Hong Kong, Hong Kong, China. [21]Comparative Experimental Pathology, Technische Universität München, Munich 81675, Germany. [22]Center of Life Sciences, Skolkovo Institute of Science and Technology, Moscow 121205, Russia. [23]Center of Molecular Medicine, CEITEC, Masaryk University, Brno 601 77, Czech Republic. [24]Hematology and Oncology–Campus Benjamin Franklin (CBF), Charité—Universitätsmedizin Berlin, Berlin 12203, Germany. [25]Wellcome Trust-MRC Stem Cell Institute, Cambridge Biomedical Campus, University of Cambridge, CB2 0XY, Cambridge, UK. [26]Department of Haematology, Cambridge University Hospitals NHS Trust, Cambridge CB2 0PT, UK. [27]Instituto de Medicina Oncológica y Molecular de Asturias (IMOMA), Oviedo 33193, Spain. [28]Departamento de Bioquímica y Biología Molecular, Facultad de Medicina, Instituto Universitario de Oncología (IUOPA), Universidad de Oviedo, Oviedo 33006, Spain. These authors contributed equally: Julia Weber, Jorge de la Rosa, Carolyn S. Grove. These authors jointly supervised this work: George S. Vassiliou, Juan Cadiñanos, Allan Bradley, Roland Rad.

