## [Peer Review File · Nature Communications]

Reviewers' comments:

Reviewer #1 (Remarks to the Author):

Weber and colleagues report the development of an *in vivo* screening platform using a novel PB transposition system to identify potential tumor suppressor genes in B-cell tumors generated in Bloom-deficient mice. This is a well written manuscript contributed by high-quality investigators. They found that 26% of CIS genes in the mouse lymphomas are recurrently mutated in human DLBCL, therefore demonstrating the validity of the system. The findings in the initial screening are confirmed and expanded in an Emu-Myc CRISP/Cas9-based system. Some findings in mice are further investigated in human tumors.

Major issues

1. One major concern is the lack of an appropriate histopathological characterization of the mouse B-cell tumors, according to WHO classification, which precludes the adequate integration of mouse and human genetic data. Each of the major mature non-Hodgkin lymphomas show distinct patterns of genetic changes: i.e. mutations activating B-cell receptor/NF- κ B signaling and blocking terminal B-cell differentiation in DLBCL of ABL subtype, whereas GCB tumors frequently show genetic changes in apoptosis (BCL2), chromatin remodeling/epigenetics (EZH2, CREBBP etc) and GNA13/related genes, which can also be extended to FCL, Burkitt and MZL tumors. In this work, mouse BCLs resemble DLBCL rather than MZL or other mature B-cell malignancies, but neither a detailed examination nor an adequate classification of the tumors (i.e ABC vs GCB) is performed (RNA expression studies, detailed histopathological examination including IHCs, IGVH clonality, etc). These studies would be necessary to further demonstrate the validity of the system, and to determine the role of the mouse CIS in human B-cell lymphoma biology.
2. Based on this, comparison of the CIS identified in the mouse lymphomas should be re-examined in databases from patients with the corresponding lymphoma subtype (i.e. DLBCL of ABC or GCB subtype – i.e. in ref 4, with 1001 DLBCL samples from ABC or GCB subtypes). Of note, CIS in the mouse BCLs target typically mutated genes in GCB (Gna13, Ezh2, Crebbp) but also in ABCs (Foxp1, Malt1) – where those CIS co-existing in the same tumor, or alternatively the mice developed ABC and GCB lymphomas with different CIS?. In this line, genomic amplification of 2p14-16 is a recurrent change in DLBCL, frequently of the GCB subtype: where mouse lymphomas with 2p amplification of GCB subtype?.
3. An adequate histopathological classification of the mouse lymphomas will also be important to demonstrate the potential implication of the novel non-mutated genes (i.e. Phip, Rfx7, Nr3c1) in human DLBCL pathogenesis. In fig 3, a volcano plot shows downregulation of RFX7 and PHIP genes from a previously published study comparing DLBCL vs normal GC lymphocytes. However, the GSE12453 corresponds to Hodgkin lymphoma (Brune et al, JEM 2008). Besides this, expression of PHIP, RFX7, NR3C1 genes should be evaluated in the corresponding DLBCL subtype (ABC or GCB) in the several databases available (Reddy Cell 2017, Schmitz NEJM 2018, Chapuy Nat Med 2018, among others). These would allow correlations between the deregulation of these non-mutated genes with biological/clinical features in the patients.
4. Authors mention that rare truncating mutations in RFX7 are found in human BCLs according to refs 8 and 52, which correspond to two Burkitt lymphoma studies. Can these be shown? Burkitt lymphoma is a germinal-center B-cell malignancy, how the Rfx7-derived mouse lymphomas were classified? Further, rare truncating mutations in PHIP (according to refs 8, 10 and 55) are found in BCLs (DLBCL and BL papers). Were those Phip mouse tumors of GCB origin?
5. One major conclusion of the study is the identification of tumor suppressor genes in mouse B-cell lymphomas, a role that is demonstrated for Phip and Rfx7 genes in the mouse tumors. However, such role is not addressed in human tumors. Are these genes downregulated in primary ABC vs GCB biopsies/cell lines vs normal B lymphocytes by PCR/IHC? Does shRNA/sgRNA downregulation of such genes in DLBCL cell lines induce a phenotype that supports their TSG role? These functional studies would strength the clinical and scientific impact of this work.

Minor changes

Were mouse tumors clonal with respect to IGVH genes?

Ref 10 could be replaced by any of the two previous MZL studies published in JEM (Rossi et al; Kiel et al, JEM 2012)

Page 3, AML, ALL, CML, CLL: need definition

Page 5, "In contrast to human BCL samples, which often show multiple copy number alterations (ref 45), recurrently amplified/deleted regions in the murine BCL cases were rare. What type of human BCL?

Ref 50 is missing

Reviewer #2 (Remarks to the Author):

In this manuscript by Weber et al., the authors report on the development of PiggyBac transposon-mediated genetic tools and mouse models for tumor-suppressor screening and successfully demonstrate the application of these tools to study B cell lymphomagenesis. In particular, the authors use a hypomorphic Bim allele in conjunction with a novel inactivating PB transposon system in mice to achieve genome-wide TSG screening in BCL. For validation purposes, they also generate a novel Rosa-targeted Cas9.HA allele. This work revealed numerous hits, involved in diverse molecular pathways, including chromatin- and transcriptional regulators, signaling mediators and regulators of RNA metabolism. They functionally validate Rfx7 and Phip, which they demonstrate to act as tumor suppressors in lymphomagenesis in vivo. The tumor-suppressive role of these two genes in the context of B cell lymphomagenesis is new and interesting to the colleagues working in the field.

Overall, this is a very solid and interesting piece of work and the manuscript is well-written. I only have a few comments.

1.) The comparison between their top CISs and the dataset reported by Reddy et al. is interesting. However, the Reddy dataset focuses on DLBCL, whereas the B cell lymphomas analyzed here may not be restricted to this sub-entity. In fact, the authors even show evidence that some of their lymphomas display signs of plasma cell differentiation (Fig. S2). Are those plasmablastic lymphomas? It would be very informative, if the authors could provide a more detailed analyses of the lymphomas that they are isolating from their mice. They state that the lymphomas were almost exclusively DLBCL. How do they distinguish between DLBCL, Burkitt's lymphoma and plasmablastic lymphoma? Did they find and lymphoblastic lymphoma? Was the degree of bone marrow infiltration analyzed? What is the percentage of plasmacytoid and plasma cell malignancies? With this information in hand, the authors could then go back the published literature and compare the entity-specific CISs with the corresponding human datasets.

3.) Are the lesions that were analyzed clonal, or may some of the lesions simply be lymphoproliferative lesions? Can the authors provide proof that the lymphomas that were analyzed are of clonal origin?

2.) The reported data set is very interesting. I wonder whether the authors could provide a statement with regards to the possible druggability. It would be very informative for the readers with a clinical background to understand whether the hits that they identify may be associated with actionability.

Reviewer #3 (Remarks to the Author):

PiggyBac transposon tools for tumor suppressor screening identify B-cell lymphoma drivers in mice

Overall summary of paper, incl. study design

In this study, Weber et al use inactivating piggyBac mutagenesis to identify cancer genes in B-cell Lymphoma (BCL). The rationale here is that although systematic sequencing efforts have identified many of the genomically altered genes complicit in tumorigenesis, forward genetic screens can identify additional non-mutated genes also involved. The groups of Rad and Bradley have a deep expertise in mouse models of cancer and transposon-based mutagenesis for cancer gene discovery. They propose overcoming the difficulty in generating BCL using transposon mutagenesis in previous studies by combining the Blmm3/m3 allele with a novel inactivating PB transposon system in mice to achieve genome-wide TSG screening in BCL. A large proportion of the mice (>2/3) developed haematological malignancies and among these almost 90% were BCLs. Importantly, the majority of these murine BCLs harboured copy number gains in a region recurrently amplified in human BCL and containing the canonical BCL oncogenes Bcl11a and Rel. Importantly, of the top 50 most significant CIS genes identified, 22 were known BCL cancer genes or present in the COSMIC cancer gene census. Additionally, the top 8 such genes are all known BCL genes in humans. Significantly, most (74%) of these CIS genes are not recurrently mutated in clinical BCL sequencing studies, highlighting the importance of genome-wide genetic screens in their identification. Indeed, analysis of expression data from clinical BCL samples confirmed an enrichment for these genes to be transcriptionally silent.

They elected to validate two novel CIS genes in particular, Rfx7 and Phip. Both genes were substantially downregulated in human DLBCL as compared to non-malignant B cells and are involved in transcriptional regulation. The CRISPR/Cas9 in vivo system used (ex vivo lentiviral transduction of sgRNA followed by transplantation in mice) appears to be an efficient means to gene silencing (20% efficiency) and the confirmation using Trp53 is compelling. Although neither gene is recurrently genomically altered in human BCL, silencing of these genes in the mouse models confirmed likely role as tumour suppressor genes. Although beyond the scope of this paper, some discussion about whether CRISPR knockout screens could be used to identify novel vulnerabilities in BCL models with these TSGs would be welcome.

Finally, using gene expression & survival data from a large clinically annotated DLBCL patient cohort the authors demonstrated that low expression of 11 of the 50 CIS genes was associated with decreased survival.

Overall, this is a robust, well-designed experiment from a group with considerable expertise in this area. It identifies novel tumour suppressor genes in BCL and points to a next wave of experiments to define how best to identify vulnerabilities in BCL cells harbouring these events. It is well suited for the readership of this journal and will be of considerable interest not only to experimentalists in haem malignancies. I recommend for publication with no major criticisms.

Specific comments

1. Cas9 expression in murine models is typically antigenic and leads to rejection of expressing cells – did the authors consider using an inducible expression system to allow tight on-off control of the HSPC cells ex-vivo for sgRNA gene targeting instead?
2. Did the authors consider more formal pathway or network analysis of their 50 CIS genes (IPA, GSEA, pathway commons etc) to identify the range of biology underpinning these loss-of-function events?

Reviewer #4 (Remarks to the Author):

In this manuscript, Weber et al. describe experiments designed to determine if a PiggyBac transposon-based forward genetic screen could be used for identifying genetic drivers of B cell lymphoma (BCL), specifically new tumor. They further describe a new in vivo validation approach of candidate BCL driver genetic alterations. Moreover, the manuscript also describes new and useful loss-of-function transposon variants that can be mobilized by PiggyBac or Sleeping Beauty transposases. The splice acceptors (SA) used in these vectors were tested in the Hprt locus and the transposon vectors, and transgenic mouse lines produced that carry them, could be useful for other cancer gene screens in the future. Moreover, the screens described here were done on a homozygous *Blim* gene (*Blm*) hypomorphic mutant background that both predisposes to BCL development, and increases the rate of loss of heterozygosity (LOH), which should increase the chance of TSG recovery. This manuscript is somewhat unique in the field of mouse somatic cell transposon mutagenesis in the sense that it uses transposon vectors specifically designed for loss of function screening for TSGs. This manuscript presents a large body of work that, if presented with more scientific support and data-driven conclusions would be a valuable addition to the lymphoma literature. However, the current manuscript is lacking in scientific/data support for some of the major conclusions of the paper. The models themselves are superficially examined. This new model of BCL could be an amazing tool for many labs to use. It is critical the authors take care in providing a very detailed description of the models/results. Specific comments below:

1. The authors should add a statement of rationale for focusing only on the identification of tumour suppressor genes (TSGs). E.g. Are most known genes involved in DLBCL to date TSGs?
2. With the model system, tumor suppressor gene haploinsufficiency would be unveiled. Do the authors find any evidence of this occurring? For example, with *Pten* as it has been described as haploinsufficient for tumor suppression in some settings? Related to this, can the authors show that insertion mutations in TSG are reduced to homozygous state by the *Blm*^{3m/3m} background, as compared to other screens not done on this background?
3. The authors discussed embryonic lethality in their models. Would they please elaborate on the data that supports this observation that BCLs develop specifically? Number of litters birthed, number of animals in each genotype per litter. This should be data they already gathered, but important to include to make that statement. In addition, could the lethality be specific to a the *Blm*^{m3/m3} background? Do *ITP1-C; Rosa26PB/+* mice survive?
4. Do double transgenic mice (*ITP1-C; Rosa26PB/+* or *ITP2-M; Rosa26PB/+*) develop any phenotypes? It is very important to include data on that genotype and the *Blm*^{3m/3m} alleles alone.
5. With the major claim being that this is a new model for B cell lymphoma, the authors present very little data (only supplemental Figure 2 and supplemental table 1) that is convincing that they have a B cell lymphoma phenotype. The authors should elaborate on how their pathologists scored the tissues, what tissues they assessed, and if any additional histological markers were used. In addition, have the authors performed any flow cytometric analyses on the samples to better characterize the lymphomas? Was *Igh* or *Igk* rearrangement data obtained? Please provide more evidence to support this major claim.
6. Related to the issue of *Ig* gene rearrangement, is the disease presented oligoclonal or does each animal have a multiple lymphoid disease being driven by unique clones?
7. Related to the analysis of common insertion sites (CIS):
 - a. What about using additional methods to analyse the data to more comprehensively identify CIS genes (other statistical methods?)
 - b. Are any co-occurring CIS observed?
8. The authors comment that 29% of animals developed various solid cancers. What are they? Could some of these actually been lymphoid disease invading other tissues? A supplemental figure/table describing these observations is needed to have a fully comprehensive view of the model.
9. The authors can refer to Moriarity et al. 2015 Nature Genetics to support their hypothesis about transposon-driven mutagenesis is a key cancer-promoting factor that can drive tumours in absence of chromosomal rearrangements. Similar observations were made in an osteosarcoma model using SB.
10. With regard to comparisons of mouse BCL CIS-associated genes to human DLBCL genomic

alterations, the authors used gene expression levels but an analysis of gene copy number loss would be useful to include. That is, how many BCL CIS-associated genes show copy number loss or gain in human DLBCL. The authors speculate that some of the BCL CIS-associated genes are lost in human DLBCL rather than suffering obvious loss of function mutations.

11. The Crispr/Cas9 based in vivo model presented is a powerful system to validate genetic drivers for many hematopoietic diseases. The authors need to provide more evidence (histology, flow) describing the FLC populations targeted for modification and the resulting phenotypes. No data are shown to support lymphoma phenotypes. Moreover, the lymphomas induced by loss of Phip and Rfx7 should be analysed at the level of protein, that is, do the lymphomas produced express reduced or no protein for the target gene.

12. When the author states “strikingly high number of genes involved in....” please put in the % of genes that fall into your descriptors and whether enrichment is statistically significant?

Suggested changes to figures:

- Figure 1:

- o include the evidence that disease is really BCL

- o include the double transgenic animals on survival curves

- o embryonic lethality bar graph doesn't add much and can just be mentioned in text

- Figure 2:

- o The cellular localization data is really glossed over in the text and therefore is not really necessary. If authors highlight specifically a gene with unknown function and make suggestions about it, then it would be worth keeping.

- Figure 3:

- o Could be lumped in with Figure 2, doesn't need to be stand alone.

- Figure 4:

- o Provide evidence that animals are developing BCL

- Figure 5:

- o Can be combined with figure 4 and then performing similar analysis on Rfx7 as well.

- Figure 6:

- o Although interesting, the figure itself is underwhelming. Perhaps focus on NR3C1 that is highlighted in the discussion with including more data on this particular pathway. Include data on statements made about mutations, etc....

- o The other survival curves could be placed in a supplemental figure.

Weber, de la Rosa, Grove et al. “PiggyBac transposon tools for tumor suppressor screening identify B-cell lymphoma drivers in mice”

Point by point responses to the reviewer’s comments

We thank the reviewers for their comments and suggestions, which have helped to improve the quality of the study. We have performed a large set of new experiments and analyses, which allowed us to address all questions raised.

The data have been included into existing figures/tables and – in addition – into:

- **3 new main figures,**
- **13 novel supplementary figures and**
- **11 new supplementary tables.**

Major aspects characterized and added are:

1. Extensive analyses to **systematically characterize and classify the hematopoietic tumors** arising from the screen, including histology, immunohistochemistry, gene expression profiling, B-cell receptor repertoire sequencing, clonality analyses and others. This allowed us to provide several levels of evidence that the cancers developing in the animals are **full-blown malignant diffuse large B-cell lymphomas** reminiscent of the human disease.
2. Thorough characterization of **solid tumors** emerging from the screen
3. Comprehensive data on the **CRISPR/Cas9-based *in vivo* validation platform** (flow cytometry, histology, immunohistochemistry, expression analysis)
4. Multiple analyses to study **loss of heterozygosity** in the tumors, including NGS-based SNP profiling, immunohistochemistry, CNV analyses, etc. We demonstrate that owing to the *Blm*^{m3/m3} background of the screen, loss of heterozygosity rates are increased at tumor suppressor genes, which can facilitate recessive screening.

Reviewer #1 (Remarks to the Author):

Weber and colleagues report the development of an in vivo screening platform using a novel PB transposition system to identify potential tumor suppressor genes in B-cell tumors generated in Bloom-deficient mice. This is a well written manuscript contributed by high-quality investigators. They found that 26% of CIS genes in the mouse lymphomas are recurrently mutated in human DLBCL, therefore demonstrating the validity of the system. The findings in the initial screening are confirmed and expanded in an Emu-Myc CRISP/Cas9-based system. Some findings in mice are further investigated in human tumors.

Major issues

1. One major concern is the lack of an appropriate histopathological characterization of the mouse B-cell tumors, according to WHO classification, which precludes the adequate integration of mouse and human genetic data. Each of the major mature non-Hodgking lymphomas show distinct patterns of genetic changes: i.e. mutations activating B-cell receptor/NF- κ B signaling and blocking terminal B-cell differentiation in DLBCL of ABL subtype, whereas GCB tumors frequently show genetic changes in apoptosis (BCL2), chromatin remodeling/epigenetics (EZH2, CREBBP etc) and GNA13/related genes, which can also be extended to FCL, Burkitt and MZL tumors. In this work, mouse BCLs resemble DLBCL rather than MZL or other mature B-cell malignancies, but neither a detailed examination nor an adequate classification of the tumors (i.e ABC vs GCB) is performed (RNA expression studies, detailed histopathological examination including IHCs, IGVH clonality, etc). These studies would be necessary to further demonstrate the validity of the system, and to determine the role of the mouse CIS in human B-cell lymphoma biology.

As suggest by the reviewer, we now systematically characterized the B-cell lymphomas emerging from this screen in detail:

Histopathology/Immunohistochemistry

We first performed a detailed histopathological classification of the lymphomas based on tumor morphology and marker expression (immunohistochemistry). All analyses were conducted by Leticia Quintanilla-Martinez, who has long-standing expertise in mouse hematopathology.

We characterized 59 hematopoietic tumors using an immunohistochemistry (IHC) panel comprising the markers B220 (specific for B cells), CD3 (T cells), myeloperoxidase (myeloid cells) and CD138 (plasma cells). The vast majority of tumors were B cell neoplasms (52/59; 88.1%). Only six CD3 positive T-cell lymphomas (10.2%) and one tumor with myeloid differentiation (1.7%) were found.

B-cell lymphomas were almost exclusively reminiscent of human diffuse large B-cell lymphoma (DLBCL; 51/52; 98.1%). Neoplasms usually manifested in mesenteric lymph nodes and/or spleens, with moderate or extensive alterations of lymphoid organ architecture due to abnormal B-cell expansion (shown by B220 IHC). DLBCLs were composed of large-sized neoplastic cells (centroblasts) with abundant cytoplasm, a round nucleus, vesicular chromatin, with two or more

nucleoli, which were often membrane-bound, and showed high proliferation rates (as demonstrated by Ki-67 IHC).

In all tumors, we also observed a small percentage of lymphoid cells with immunoblastic morphology (larger cells with abundant cytoplasm and one prominent centrally localized nucleolus).

A subset of DLBCL cases showed characteristics of plasmacytic differentiation (13/51; 25.5%), in which a subset of tumor cells lost the B cell marker (B220) and expressed the plasma cell marker (CD138). A significant proportion of tumor cells retained however B220 expression, distinguishing these cancers from plasmablastic lymphoma or other plasma cell malignancies. In general, tumor cell infiltration into organs located within the thoracic and abdominal cavities such as lungs, liver, intestine and kidneys was frequently observed (37/42 analyzed DLBCLs; 88.1%) while bone marrow infiltration was rare (2/42; 4.8%).

To sub-classify the DLBCL cases based on their cell of origin, we performed IHC using the germinal center marker Bcl6 and the post-germinal center marker Mum1/Irf4. Expression of MUM1/IRF4 is associated with non-GCB DLBCL in humans. In contrast, BCL6 expression is primarily associated with germinal center B-cell like- (GCB) DLBCL, although a subset of non-GCB DLBCL is also positive for BCL6. We analyzed 20 samples and found that 15 cases (75%) were Bcl6-positive/Irf4-negative, suggesting a GCB DLBCL phenotype. Five cases (25%) were Irf4-positive/Bcl6-negative, which we classified as non-GCB DLBCL.

The results are shown in Figure 2, Figure S3, Figure S4 and Table S4 and are referred to on pages 6 and 7 of the manuscript.

Gene expression profiling

In addition, we performed RNA sequencing of DLBCL samples ($n = 25$) for gene-expression profiling (GEP), which is considered the gold standard for DLBCL sub-classification in humans. Using the murine orthologues of the human classifier genes, mouse DLBCLs clustered into two main clusters. Cluster B contained exclusively IHC-diagnosed GCB tumors, whereas all five IHC-diagnosed non-GCB cancers fell into cluster A. Like in human DLBCL, IHC-based and GEP-based tumor classification are not fully superimposable. In fact, the discordance in mice might be even stronger because mouse DLBCLs are less “homogeneous”: we observed that GCB tumors often contain infiltrates of CD3 positive T lymphocytes and residual plasma cells, which makes accurate GEP from whole tumor lysates challenging. This might be a reason why some of the IHC-diagnosed GCB samples fall into cluster A. Taken together these data show mouse DLBCLs can be sub-classified similarly to human DLBCLs. Thus, our model recapitulates key aspects of the human disease.

The results can be found in Figure 2 and are referred to on page 7, 3rd paragraph of the manuscript.

Clonality

For clonality analysis, we performed RNA-based immunoglobulin repertoire profiling of 30 DLBCL cases. To this end, we conducted full-length amplification of the variable regions of the immunoglobulin heavy and light chains. To eliminate PCR and sequencing errors leading to incorrect clone assignments, unique molecular identifiers (UMI) were introduced during cDNA synthesis. Immune repertoires were sequenced on Illumina MiSeq. For data analysis, de-multiplexing and UMI consensus sequence assembly was performed with *MIGEC*. *MiXCR* was used for clone detection based on the highly variable complementarity-determining region 3 (CDR3). To visualize the clonal structure of tumors, we developed a custom script to generate clonality network plots. In these plots, each clone constitutes a node of the network. The size of the node correlates with the number of reads assigned to it, and clones differing by only 1 bp in their CDR3 sequence are linked. The complexity of the branching (i.e. number of subclones) is a measure for the grade of somatic hypermutation (SHM), which is a hallmark of DLBCL. We highlighted clones that consist of more than 10% of the total reads of a sample in color (red, blue or green). Different clones (as defined by a unique V(D)J rearrangement) are marked with different colors. Note that RNA was isolated from whole tumor tissue lysates that contain varying amounts of non-transformed B cells (most likely accounting for the small gray nodes in the plots).

The vast majority of tumors (16/27) were monoclonal, indicating that these were full-blown malignant lymphomas arising from one transformed B cell. Eight samples consisted of two dominant clones, suggesting the presence of two independent malignant DLBCLs in one mouse. Only two tumors arose from multiple clones. We found evidence of SHM in the majority of tumors. As expected, there were differences in the extent of SHM between individual tumors, and between heavy and light chains of the same clone, as these undergo SHM separately.

The results are shown in Figure 3, Figure S5, Figure S6, Figure S7, Figure S8 and Table S5 and are referred to on pages 7 and 8 of the manuscript.

2. Based on this, comparison of the CIS identified in the mouse lymphomas should be re-examined in databases from patients with the corresponding lymphoma subtype (i.e. DLBCL of ABC or GCB subtype – i.e. in ref 4, with 1001 DLBCL samples from ABC or GCB subtypes). Of note, CIS in the mouse BCLs target typically mutated genes in GCB (Gna13, Ezh2, Crebbp) but also in ABCs (Foxp1, Malt1) – where those CIS co-existing in the same tumor, or alternatively the mice developed ABC and GCB lymphomas with different CIS?. In this line, genomic amplification of 2p14-16 is a recurrent change in DLBCL, frequently of the GCB subtype: where mouse lymphomas with 2p amplification of GCB subtype?

An in-depth systematic search for subtype-specific drivers is not possible in our cohort because of the small number of ABC DLBCL ($n = 5$). Among GCB DLBCL we frequently found transposon hits in genes typically mutated in human GCB DLBCL (e.g. *GNA13*, *CREBBP*), which in some cases co-existed with hits in genes that are more typical for ABC type human tumors (e.g. *MALT1*). Similarly, *Rel* amplifications affecting Chr11 were observed in both ABC and GCB mouse tumors,

suggesting that those genetic events are not exclusive for one or the other DLBCL subtype. This mirrors somewhat the human situation. For example, although *REL* amplifications are more frequent in human GCB DLBCL (28% of cases), they also occur in 5% of human ABC DLBCL (meta-analysis shown in Nogai *et al.* Journal of Clinical Oncology 2011). This information has been added to the legend of Figure 4.

*3. An adequate histopathological classification of the mouse lymphomas will also be important to demonstrate the potential implication of the novel non-mutated genes (i.e. *Phip*, *Rfx7*, *Nr3c1*) in human DLBCL pathogenesis. In fig 3, a volcano plot shows downregulation of *RFX7* and *PHIP* genes from a previously published study comparing DLBCL vs normal GC lymphocytes. However, the GSE12453 corresponds to Hodgkin lymphoma (Brune *et al*, JEM 2008). Besides this, expression of *PHIP*, *RFX7*, *NR3C1* genes should be evaluated in the corresponding DLBCL subtype (ABC or GCB) in the several databases available (Reddy Cell 2017, Schmitz NEJM 2018, Chapuy Nat Med 2018, among others). These would allow correlations between the deregulation of these non-mutated genes with biological/clinical features in the patients.*

The reviewer rightly points out that the GSE12453 dataset is labelled as “Origin and pathogenesis of lymphocyte-predominant Hodgkin lymphoma as revealed by global gene expression analysis” in the GEO database. Given that our volcano plot shows DLBCL/normal GC lymphocyte comparisons this could be misleading. We now specifically clarify in the methods section that this data set not only contains Hodgkin Lymphoma, but also DLBCL and normal GC lymphocytes. We exclusively used the DLBCL and normal GC lymphocyte data to generate the volcano plot shown in Figure 5. This information is provided in the methods section on page 21, 5th paragraph of the manuscript.

As suggested, we now highlight *NR3C1* in all available DLBCL/normal GC lymphocyte comparisons. See volcano plots in Figure 5 and Figure S11.

We evaluated the expression of *RFX7*, *PHIP* and *NR3C1* in all publicly accessible datasets (GSE12453, GSE12195, and GSE2350) that contain normal GC lymphocyte controls (Figure 5 and Figure S11). In all available datasets *RFX7*, *PHIP* and *NR3C1* are downregulated in DLBCL samples as compared to controls.

The more recent studies mentioned by the reviewer do not contain normal GC lymphocytes, which would be needed for a comparison tumor/normal. We used these data however to perform comparisons between ABC vs GCB subtypes. We found however no (*RFX7* and *PHIP*) or only small (*NR3C1*) expression differences between human ABC/GCB subtypes.

4. Authors mention that rare truncating mutations in *RXF7* are found in human BCLs according to refs 8 and 52, which correspond to two Burkitt lymphoma studies. Can these be shown? Burkitt lymphoma is a germinal-center B-cell malignancy, how the *Rxf7*-derived mouse lymphomas were classified? Further, rare truncating mutations in *PHIP* (according to refs 8, 10 and 55) are found in BCLs (DLBCL and BL papers). Were those *Phip* mouse tumors of GCB origin?

As suggested by the reviewer, we prepared schemes displaying mutations in *RFX7* and *PHIP* from major DLBCL and Burkitt lymphoma sequencing studies.

The results are shown in Figure S17 and referred to on page 13, 1st/2nd paragraphs of the manuscript.

Mouse DLBCLs with “high-coverage” *Rfx7* or *Phip* insertions were not associated with a specific DLBCL sub-type (GCB or ABC). Likewise, in human tumors, *RFX7* and *PHIP* mutations are found in GCB- as well as ABC-type DLBCL.

5. One major conclusion of the study is the identification of tumor suppressor genes in mouse B-cell lymphomas, a role that is demonstrated for *Phip* and *Rxf7* genes in the mouse tumors. However, such role is not addressed in human tumors. Are these genes downregulated in primary ABC vs GCB biopsies/cell lines vs normal B lymphocytes by PCR/IHC? Does shRNA/sgRNA downregulation of such genes in DLBCL cell lines induce a phenotype that supports their TSG role? These functional studies would strength the clinical and scientific impact of this work.

We evaluated the expression of *RFX7*, *PHIP* and *NR3C1* in all publicly accessible datasets (GSE12453, GSE12195, and GSE2350) that contain normal GC lymphocyte controls (Figure 5 and Figure S11). In all available datasets *RFX7*, *PHIP* and *NR3C1* are downregulated in DLBCL samples as compared to controls. Furthermore, truncating mutations in *RFX7* and *PHIP* are found in human DLBCL and Burkitt lymphoma cases further supporting a tumor-suppressive function of these genes (Figure S17).

As suggested by the reviewer, we also performed shRNA mediated *in vitro* knockdown experiments using DLBCL cancer cell lines. We used the GCB cell line HT and the ABC cell line RIVA. We cloned shRNAs targeting *RFX7* ($n = 3$) and *PHIP* ($n = 4$) as well as a scrambled control shRNAs (scr) into a lentiviral vector containing shRNA and blue fluorescent protein (BFP) expression cassettes. We infected both cell lines with the lentiviral particles (16 different set-ups) and co-cultivated transduced and non-transduced cells for 17 days. Efficient knockdown was analyzed and confirmed on day 17 by real time quantitative PCR. On day 3, 7, 10, 14, and 17, we performed flow cytometry to determine the proportion of BFP positive cells. No significant differences between *RFX7* and *PHIP* knockdown cells on the one hand and scr control cells on the other hand were observed over the course of the experiment.

We would like to stress that we have previously obtained such seemingly “negative” results in numerous previous projects attempting to validate candidate cancer genes *in vitro* by using cancer cell lines from various cancer types. We therefore ideally always try to validate functionally newly

discovered genes using *in vivo* systems, which recapitulate the full complexity of tumor evolution starting in a non-transformed cell.

Compared to *in vivo* systems, limitations of *in vitro* approaches, especially if validation experiments are performed with full-blown cancer cells lines, include:

1. The “wrong” genetic, mutational or cell-of origin context in which a gene of interest is operative
2. The aggressive nature of full-blown cancer cells, often precluding the detection of subtle phenotypes
3. The restricted set of readouts *in vitro*, which don’t encompass the large spectrum of possible cancer-driving processes affected by a gene
4. Inherent differences between *in vitro* and *in vivo* situation (only *in vivo* a candidate gene can be tested in a fully functioning organism)

The results can be found in Figure S14 and are referred to on page 12, 1st paragraph of the manuscript.

Minor changes

Were mouse tumors clonal with respect to IGVH genes?

For clonality analysis, we performed RNA-based immunoglobulin repertoire profiling of 30 DLBCL cases. To this end, we conducted full-length amplification of the variable regions of the immunoglobulin heavy and light chains. To eliminate PCR and sequencing errors leading to incorrect clone assignments, unique molecular identifiers (UMI) were introduced during cDNA synthesis. Immune repertoires were sequenced on Illumina MiSeq. For data analysis, demultiplexing and UMI consensus sequence assembly was performed with *MIGEC*. *MiXCR* was used for clone detection based on the highly variable complementarity-determining region 3 (CDR3). To visualize the clonal structure of tumors, we developed a custom script to generate clonality network plots. In these plots, each clone constitutes a node of the network. The size of the node correlates with the number of reads assigned to it, and clones differing by only 1 bp in their CDR3 sequence are linked. The complexity of the branching (i.e. number of subclones) is a measure for the grade of somatic hypermutation (SHM), which is a hallmark of DLBCL. We highlighted clones that consist of more than 10% of the total reads of a sample in color (red, blue or green). Different clones (as defined by a unique V(D)J rearrangement) are marked with different colors. Note that RNA was isolated from whole tumor tissue lysates that contain varying amounts of non-transformed B cells (most likely accounting for the small gray nodes in the plots).

The vast majority of tumors (16/27) were monoclonal, indicating that these were full-blown malignant lymphomas arising from one transformed B cell. Eight samples consisted of two dominant clones, suggesting the presence of two independent malignant DLBCLs in one mouse. Only two tumors arose from multiple clones. We found evidence of SHM in the majority of tumors.

As expected, there were differences in the extent of SHM between individual tumors, and between heavy and light chains of the same clone, as these undergo SHM separately.

The results are shown in Figure 3, Figure S5, Figure S6, Figure S7, Figure S8 and Table S5 and are referred to on pages 7 and 8 of the manuscript.

Ref 10 could be replaced by any of the two previous MZL studies published in JEM (Rossi et al; Kiel et al, JEM 2012)

We replaced the reference (Lohr et al. PNAS 2012) with Rossi et al. J Exp Med 2012.

Page 3, AML, ALL, CML, CLL: need definition

We changed this in the manuscript to acute and chronic leukemia (page 4, 2nd paragraph).

Page 5, "In contrast to human BCL samples, which often show multiple copy number alterations (ref 45), recurrently amplified/deleted regions in the murine BCL cases were rare. What type of human BCL?"

We now refer in the manuscript to DLBCL as a BCL entity that often shows multiple copy number alterations (page 8, 2nd paragraph).

Ref 50 is missing

We now added the reference (Adams et al. Nature 1985).

Reviewer #2 (Remarks to the Author):

In this manuscript by Weber et al., the authors report on the development of PiggyBac transposon-mediated genetic tools and mouse models for tumor-suppressor screening and successfully demonstrate the application of these tools to study B cell lymphomagenesis. In particular, the authors use a hypomorphic Blm allele in conjunction with a novel inactivating PB transposon system in mice to achieve genome-wide TSG screening in BCL. For validation purposes, they also generate a novel Rosa-targeted Cas9.HA allele. This work revealed numerous hits, involved in diverse molecular pathways, including chromatin- and transcriptional regulators, signaling mediators and regulators of RNA metabolism. They functionally validate Rfx7 and Phip, which they demonstrate to act as tumor suppressors in lymphomagenesis in vivo. The tumor-suppressive role of these two genes in the context of B cell lymphomagenesis is new and interesting to the colleagues working in the field. Overall, this is a very solid and interesting piece of work and the manuscript is well-written. I only have a few comments.

1.) The comparison between their top CISs and the dataset reported by Reddy et al. is interesting. However, the Reddy dataset focuses on DLBCL, whereas the B cell lymphomas analyzed here may not be restricted to this sub-entity. In fact, the authors even show evidence that some of their lymphomas display signs of plasma cell differentiation (Fig. S2). Are those plasmablastic lymphomas? What is the percentage of plasmacytoid and plasma cell malignancies?

In this study, no plasma cell malignancies were observed. However, a subset of DLBCL cases showed characteristics of plasmacytic differentiation (DLBCLs with plasmacytic differentiation; 13/51; 25.5%), in which a fraction of tumor cells lost the B cell marker (B220) and expressed the plasma cell marker (CD138). A significant proportion of tumor cells in DLBCL with plasmacytic differentiation however retained expression of B cell markers such as B220 and only part of the infiltrate (10-20%) were B220 negative/CD138 positive cells. In contrast, the characteristic immunophenotype of plasmablastic lymphoma is that the neoplastic cells are negative for B cell markers but are positive for markers of plasmacytic differentiation such as CD138.

We included microscopic images showing H&E and B220 as well CD138 IHCs of DLBCLs with plasmacytic differentiation in Figure S3. We refer to the data on page 7, 1st paragraph of the manuscript.

It would be very informative, if the authors could provide a more detailed analyses of the lymphomas that they are isolating from their mice. They state that the lymphomas were almost exclusively DLBCL. How do they distinguish between DLBCL, Burkitt's lymphoma and plasmablastic lymphoma?

We performed a detailed histopathological classification of the lymphomas based on tumor morphology and marker expression (immunohistochemistry). All analyses were conducted by Leticia Quintanilla-Martinez, who has long-standing expertise in mouse hematopathology.

We characterized 59 hematopoietic tumors using an immunohistochemistry (IHC) panel comprising the markers B220 (specific for B cells), CD3 (T cells), myeloperoxidase (myeloid cells) and CD138 (plasma cells). The vast majority of tumors were B cell neoplasms (52/59; 88.1%). Only six CD3 positive T-cell lymphomas (10.2%) and one tumor with myeloid differentiation (1.7%) were found.

B-cell lymphomas were almost exclusively reminiscent of human DLBCL (51/52; 98.1%). Neoplasms usually manifested in mesenteric lymph nodes and/or spleens, with moderate or extensive alterations of lymphoid organ architecture due to abnormal B-cell expansion (shown by B220 IHC). DLBCLs were composed of large-sized neoplastic cells (centroblasts) with abundant cytoplasm, a round nucleus, vesicular chromatin, with two or more nucleoli, which were often membrane-bound, and showed high proliferation rates (as demonstrated by Ki-67 IHC).

In all tumors, we also observed a small percentage of lymphoid cells with immunoblastic morphology (larger cells with abundant cytoplasm and one prominent centrally localized nucleolus).

In contrast, Burkitt lymphomas show a characteristic morphology of medium-sized monotonous cells with blastic chromatin and inconspicuous nucleoli, starry sky pattern and high levels of apoptosis caused by *Myc* over-expression in the tumor cells. None of these features was found in the tumors of this study.

A subset of DLBCL cases showed characteristics of plasmacytic differentiation (13/51; 25.5%), in which a subset of tumor cells lost the B cell marker (B220) and expressed the plasma cell marker (CD138). A significant proportion of tumor cells retained however B220 expression, distinguishing these cancers from plasmablastic lymphoma or other plasma cell malignancies. In general, tumor cell infiltration into organs located within the thoracic and abdominal cavities such as lungs, liver, intestine and kidneys was frequently observed (37/42 analyzed DLBCLs; 88.1%) while bone marrow infiltration was rare (2/42; 4.8%).

To sub-classify the DLBCL cases based on their cell of origin, we performed IHC using the germinal center marker Bcl6 and the post-germinal center marker Mum1/Irf4. Expression of MUM1/IRF4 is associated with non-GCB DLBCL in humans. In contrast, BCL6 expression is primarily associated with germinal center B-cell like- (GCB) DLBCL, although a subset of non-GCB DLBCL is also positive for BCL6. We analyzed 20 samples and found that 15 cases (75%) were Bcl6-positive/Irf4-negative, suggesting a GCB DLBCL phenotype. Five cases (25%) were Irf4-positive/Bcl6-negative, which we classified as non-GCB DLBCL.

The results are shown in Figure 2, Figure S3, Figure S4 and Table S4 and are referred to on pages 6 and 7 of the manuscript.

Did they find and lymphoblastic lymphoma?

In this study, we observed one mouse that developed both, DLBCL and B-cell lymphoblastic lymphoma, one mouse that developed both, DLBCL and T-cell lymphoblastic lymphoma, and six animals, which showed T-cell lymphoblastic lymphoma.

The data are shown in Table S4.

Was the degree of bone marrow infiltration analyzed?

We analyzed bone marrow infiltration in all ($n = 42$) DLBCL cases. Two mice presented infiltrating tumor cells in the bone marrow.

Results are shown in Figure S4 and referred to on page 7, 1st paragraph of the manuscript.

With this information in hand, the authors could then go back the published literature and compare the entity-specific CISs with the corresponding human datasets.

An in-depth systematic search for subtype-specific drivers is not possible in our cohort because of the small number of ABC DLBCL ($n = 5$). Please see also comment to question 2 raised by reviewer 1 (page 4 of this response letter).

3.) Are the lesions that were analyzed clonal, or may some of the lesions simply be lymphoproliferative lesions? Can the authors provide proof that the lymphomas that were analyzed are of clonal origin?

For clonality analysis, we performed RNA-based immunoglobulin repertoire profiling of 30 DLBCL cases. To this end, we conducted full-length amplification of the variable regions of the immunoglobulin heavy and light chains. To eliminate PCR and sequencing errors leading to incorrect clone assignments, unique molecular identifiers (UMI) were introduced during cDNA synthesis. Immune repertoires were sequenced on Illumina MiSeq. For data analysis, de-multiplexing and UMI consensus sequence assembly was performed with *MIGEC*. *MiXCR* was used for clone detection based on the highly variable complementarity-determining region 3

(CDR3). To visualize the clonal structure of tumors, we developed a custom script to generate clonality network plots. In these plots, each clone constitutes a node of the network. The size of the node correlates with the number of reads assigned to it, and clones differing by only 1 bp in their CDR3 sequence are linked. The complexity of the branching (i.e. number of subclones) is a measure for the grade of somatic hypermutation (SHM), which is a hallmark of DLBCL. We highlighted clones that consist of more than 10% of the total reads of a sample in color (red, blue or green). Different clones (as defined by a unique V(D)J rearrangement) are marked with different colors. Note that RNA was isolated from whole tumor tissue lysates that contain varying amounts of non-transformed B cells (most likely accounting for the small gray nodes in the plots).

The vast majority of tumors (16/27) were monoclonal, indicating that these were full-blown malignant lymphomas arising from one transformed B cell. Eight samples consisted of two dominant clones, suggesting the presence of two independent malignant DLBCLs in one mouse. Only two tumors arose from multiple clones. We found evidence of SHM in the majority of tumors. As expected, there were differences in the extent of SHM between individual tumors, and between heavy and light chains of the same clone, as these undergo SHM separately.

The results are shown in Figure 3, Figure S5, Figure S6, Figure S7, Figure S8 and Table S5 and are referred to on pages 7 and 8 of the manuscript.

2.) The reported data set is very interesting. I wonder whether the authors could provide a statement with regards to the possible druggability. It would be very informative for the readers with a clinical background to understand whether the hits that they identify may be associated with actionability.

As suggested by the reviewer, we analyzed the potential druggability of the human orthologues of the top 50 list mining the Drug Gene Interaction Database.

The results can be found in Table S11 and are referred to on page 14, 2nd paragraph of the manuscript.

Reviewer #3 (Remarks to the Author):

PiggyBac transposon tools for tumor suppressor screening identify B-cell lymphoma drivers in mice

Overall summary of paper, incl. study design

In this study, Weber et al use inactivating piggyBac mutagenesis to identify cancer genes in B-cell Lymphoma (BCL). The rationale here is that although systematic sequencing efforts have identified many of the genomically altered genes complicit in tumorigenesis, forward genetic screens can identify additional non-mutated genes also involved. The groups of Rad and Bradley have a deep expertise in mouse models of cancer and transposon-based mutagenesis for cancer gene discovery. They propose overcoming the difficulty in generating BCL using transposon mutagenesis in previous studies by combining the Blmm3/m3 allele with a novel inactivating PB transposon system in mice to achieve genome-wide TSG screening in BCL. A large proportion of the mice (>2/3) developed haematological malignancies and among these almost 90% were BCLs. Importantly, the majority of these murine BCLs harboured copy number gains in a region recurrently amplified in human BCL and containing the canonical BCL oncogenes Bcl11a and Rel. Importantly, of the top 50 most significant CIS genes identified, 22 were known BCL cancer genes or present in the COSMIC cancer gene census. Additionally, the top 8 such genes are all known BCL genes in humans. Significantly, most (74%) of these CIS genes are not recurrently mutated in clinical BCL sequencing studies, highlighting the importance of genome-wide genetic screens in their identification. Indeed, analysis of expression data from clinical BCL samples confirmed an enrichment for these genes to be transcriptionally silent.

They elected to validate two novel CIS genes in particular, Rfx7 and Phip. Both genes were substantially downregulated in human DLBCL as compared to non-malignant B cells and are involved in transcriptional regulation. The CRISPR/Cas9 in vivo system used (ex vivo lentiviral transduction of sgRNA followed by transplantation in mice) appears to be an efficient means to gene silencing (20% efficiency) and the confirmation using Trp53 is compelling. Although neither gene is recurrently genomically altered in human BCL, silencing of these genes in the mouse models confirmed likely role as tumour suppressor genes.

Although beyond the scope of this paper, some discussion about whether CRISPR knockout screens could be used to identify novel vulnerabilities in BCL models with these TSGs would be welcome.

We agree with the reviewer that identification of vulnerabilities is of considerable interest, in particular with regard to a potential translation of our findings into the clinic. In this connection, other groups already performed CRISPR/Cas mediated dropout screens in human DLBCL cell lines and identified, for example, DLBCL sub-type (i.e. GCB and ABC) specific vulnerabilities (Reddy et al. Cell 2017). For the future, more refined screens, e.g., analyzing dependencies in the context of major drivers or employing DLBCL *in vivo* models, are conceivable. Relating to a potential clinical relevance, we show that a subset of our identified candidate genes functions as predictive markers for human DLBCL since low expression of these genes correlates with poor

patient survival. Among them is *NR3C1*, which codes for the glucocorticoid receptor, a “druggable gene” as the glucocorticoid prednisone is part of the standard treatment regime in different hematopoietic cancer types. Furthermore, in T-cell acute lymphoblastic leukemia resistance to glucocorticoid treatment has been previously associated with single nucleotide polymorphisms or deletions in *NR3C1*. We now discuss this on page 14, 2nd paragraph of the manuscript and in note of Table S11, in which we further list potentially druggable genes.

Finally, using gene expression & survival data from a large clinically annotated DLBCL patient cohort the authors demonstrated that low expression of 11 of the 50 CIS genes was associated with decreased survival.

Overall, this is a robust, well-designed experiment from a group with considerable expertise in this area. It identifies novel tumour suppressor genes in BCL and points to a next wave of experiments to define how best to identify vulnerabilities in BCL cells harbouring these events. It is well suited for the readership of this journal and will be of considerable interest not only to experimentalists in haem malignancies. I recommend for publication with no major criticisms.

Specific comments

1. Cas9 expression in murine models is typically antigenic and leads to rejection of expressing cells – did the authors consider using an inducible expression system to allow tight on-off control of the HSPC cells ex-vivo for sgRNA gene targeting instead?

We fully agree with the reviewer that readers need to be aware of Cas9’s possible antigenic potential in murine CRISPR/Cas models. For this reason, we now call attention to this important consideration in the methods section of the manuscript (page 23, 3rd paragraph). However, we think that immunogenicity of Cas9 is not problematic in our *in vivo* setting as we (i) transplant Cas9-expressing cells into irradiated recipient mice without host immune system and (ii) observe early on-set tumors and strong cancer phenotypes.

2. Did the authors consider more formal pathway or network analysis of their 50 CIS genes (IPA, GSEA, pathway commons etc) to identify the range of biology underpinning these loss-of-function events?

We performed enrichment analyses using the Reactome gene sets from the *MSigDB database* v6.2 and the top 50 CIS genes as an input. Among the most enriched signatures – and consistent with the role of the identified genes as DLBCL drivers – we found “Genes involved in Immune System”, “Genes involved in Adaptive Immune System” and “Genes involved in Signaling by the B Cell Receptor (BCR)”. However, as the power of this and similar statistical approaches is limited by the low input size (50 genes), we additionally performed in-depth literature search to annotate functionally the top 50 CIS genes.

The results are shown in Table S9 and Figure 4 and referred to on page 9, 4th paragraph of the manuscript.

Reviewer #4 (Remarks to the Author):

In this manuscript, Weber et al. describe experiments designed to determine if a PiggyBac transposon-based forward genetic screen could be used for identifying genetic drivers of B cell lymphoma (BCL), specifically new tumor. They further describe a new in vivo validation approach of candidate BCL driver genetic alterations. Moreover, the manuscript also describes new and useful loss-of-function transposon variants that can be mobilized by PiggyBac or Sleeping Beauty transposases. The splice acceptors (SA) used in these vectors were tested in the Hprt locus and the transposon vectors, and transgenic mouse lines produced that carry them, could be useful for other cancer gene screens in the future. Moreover, the screens described here were done on a homozygous Blooms gene (Blm) hypomorphic mutant background that both predisposes to BCL development, and increases the rate of loss of heterozygosity (LOH), which should increase the chance of TSG recovery. This manuscript is somewhat unique in the field of mouse somatic cell transposon mutagenesis in the sense that it uses transposon vectors specifically designed for loss of function screening for TSGs. This manuscript presents a large body of work that, if presented with more scientific support and data-driven conclusions would be a valuable addition to the lymphoma literature. However, the current manuscript is lacking in scientific/data support for some of the major conclusions of the paper. The models themselves are superficially examined. This new model of BCL could be an amazing tool for many labs to use. It is critical the authors take care in providing a very detailed description of the models/results. Specific comments below:

1. The authors should add a statement of rational for focusing only on the identification of tumour suppressor genes (TSGs). E.g. Are most known genes involved in DLBCL to date TSGs?

For B-cell lymphoma and DLBCL specifically, a lot of knowledge about oncogenes (such as *BCL2*, *BCL6* and *MYC*) was acquired from cytogenetic studies and retroviral insertional mutagenesis screens. In contrast, the role of TSGs in DLBCL is by far less well studied. We believe that screening is best performed in an *in vivo* setting, which recapitulates the full complexity of tumor evolution, including the sequential molecular alterations occurring in the different cellular contexts/states. Recessive screening at an organismal level still poses a major challenge. Although CRISPR/Cas based screening is potentially powerful in this respect, a lot of obstacles still have to be overcome before their efficient use *in vivo*, e.g. efficient delivery of CRISPR/Cas components to different organ types, library representation, potential immunogenicity, low editing efficiency in non-transformed cells *in vivo*. Because of these and other reasons, genome-wide CRISPR screening could so far not be achieved *in vivo*. For these reasons, we decided to develop a novel genome-wide TSG screening system using DLBCL as a prototype. We added these thoughts to the manuscript on page 4, 3rd paragraph.

2. With the model system, tumor suppressor gene haploinsufficiency would be unveiled. Do the authors find any evidence of this occurring? For example, with *Pten* as it has been described as haploinsufficient for tumor suppression in some settings? Related to this, can the authors show that insertion mutations in TSG are reduced to homozygous state by the *Blm3m/3m* background, as compared to other screens not done on this background?

To address the reviewer's question and examine whether LOH is indeed increased at TSGs hit by transposons, we analyzed the *Pten* locus, which is one of the top CIS genes in the screen. One important consideration is that insertions can be subclonal, and in such cases bulk-sequencing based LOH studies are not possible. To address this issue, we exploited the semi-quantitative nature of our insertion site sequencing approach (QiSeq), which allowed us to draw conclusions about the clonal representation of any given transposon integration. For LOH studies, we only used cancers with "high-coverage" insertions, suggesting their presence in the major tumor clone.

We first performed single nucleotide polymorphism (SNP) analysis using amplicon based NGS of ten corresponding DLBCL tissues and tail samples from *IPB* mice. In six animals, a heterozygous *Pten* germline variant was identified in the tail, allowing SNP-based LOH profiling at this locus. In three out of these six DLBCL cases, the heterozygous SNP was detected at similar variant frequencies in DLBCL tissue and tail, indicating either a lack of LOH in the tumor sample or a false-negative result due to "contaminating" non-tumor cells in the tissue. In the other three DLBCLs, variant frequencies deviated substantially from 0.5, clearly demonstrating LOH at the *Pten* locus. Thus, LOH occurred in at least 50% of those tumors, in which LOH analysis was possible. We also observed LOH at other loci, such as the *Apc* gene in an *IPB* small intestine tumor, showing that LOH is not restricted to the *Pten* locus.

To examine LOH in samples where SNP based LOH analysis is not possible (no heterozygous germline variants at the locus of interest), we also scored *Pten* expression by IHC in the 10 *IPB* DLBCLs with "high-coverage" *Pten* insertions. The vast majority of samples (8/10) had lost *Pten* expression (scored as "negative" or "weak"), indicating LOH in the tumors.

We next examined if the hypomorphic *Blm^{m3/m3}* context is associated with increased LOH rates at CIS genes. To this end, we took advantage of a similar whole-body screen, which we performed in *Blm*-proficient mice (unpublished). A critical consideration is that many tumor suppressors can act in a haploinsufficient manner, which might be cell/entity-specific. For example, it is known that *Pten* can act as a haploinsufficient TSG, but is also frequently inactivated homozygously in various cancer types. Little is known how the cellular/entity-context affects either scenario. We therefore considered it essential to look at the same entity when performing comparative LOH analyses in *Blm^{m3/m3}* and *Blm*-proficient screens. In *Blm*-proficient mice BCLs are rare, but the large size of the screen ($n = 256$ tumors) allowed us to collect a sufficient number of DLBCLs ($n = 7$) for such analyses. All seven DLBCLs had high-coverage *Pten* insertions, supporting side-by-side comparisons with corresponding *Pten*-altered cancers in the *Blm^{m3/m3}* cohort.

We performed IHC-based semi-quantitative assessment of *Pten* expression and found that the majority of *Blm*-proficient DLBCLs (5/7) expressed substantial *Pten* levels, excluding the possibility of homozygous *Pten* inactivation and LOH in these samples. In contrast, among the above described *Blm^{m3/m3}* DLBCLs only 2/10 showed evidence for a lack of LOH ($p = 0.03$, Fisher's

exact test). Altogether, these data support a model in which the $Blm^{m3/m3}$ context elevates LOH at CIS genes, thereby facilitating recessive screening.

The results can be found in Figure 6 and Figure S13 and the data are referred to on pages 10 and 11 of the manuscript.

3. The authors discussed embryonic lethality in their models. Would they please elaborate on the data that supports this observation that BCLs develop specifically? Number of litters birthed, number of animals in each genotype per litter. This should be data they already gathered, but important to include to make that statement.

As suggested, we have now added this information to the manuscript. An overview of all breedings conducted for the generation of $ITP1-C;Rosa26^{PB/+};Blm^{m3/m3}$ and $ITP2-M;Rosa26^{PB/+};Blm^{m3/m3}$ mice (with numbers of litters birthed and numbers of mice of each genotype per litter) can be found in Table S1. The data are referred to on page 6, 1st paragraph of the manuscript.

4. In addition, could the lethality be specific to a the $Blm^{m3/m3}$ background? Do $ITP1-C;Rosa26^{PB/+}$ mice survive? Do double transgenic mice ($ITP1-C;Rosa26^{PB/+}$ or $ITP2-M;Rosa26^{PB/+}$) develop any phenotypes? It is very important to include data on that genotype and the $Blm^{m3/m3}$ alleles alone.

We now clarify in the manuscript that the extensive lethality observed in the $ITP1-C;Rosa26^{PB/+};Blm^{m3/m3}$ cohort is primarily linked to the high transposon copy number of $ITP1-C$ mice. There are several layers of evidence supporting this: First, $Blm^{m3/m3}$ mice are fertile and viable and develop no embryonic phenotypes (Luo *et al.* Nature Genetics 2000). Second, we have previously observed that high-copy transposon mouse lines produce – when combined with *PiggyBac* transposase mice – extensive embryonic lethality (Rad *et al.* Science 2010; although the ATP transposon type was slightly different in that cohort). For example, double transgenic $ATP;Rosa26^{PB/+}$ with transposon copy numbers similar to $ITP1-C;Rosa26^{PB/+}$ mice (e.g., $ATP1-H8;Rosa26^{PB/+}$, $ATP1-H39;Rosa26^{PB/+}$, $ATP3-S2;Rosa26^{PB/+}$ mice) showed comparable or even higher rates of embryonic lethality in a Bloom wild-type background. We observed such transposon copy-number dependent effects on embryonic lethality across all the ~30 transposon mouse lines we looked at so far. Third, in $ITP2-M;Rosa26^{PB/+};Blm^{m3/m3}$ mice embryonic lethality is dramatically reduced as compared to $ITP1-C;Rosa26^{PB/+};Blm^{m3/m3}$ mice. $ITP1-C$ has 70 transposon copies whereas $ITP2-M$ has only 35 copies. Thus, although we have not bred $ITP1-C;Rosa26^{PB/+}$ mice without the $Blm^{m3/m3}$ allele, the above results show at multiple levels that embryonic lethality is primarily linked to the transposon copy number.

We have added this information into the methods section (pages 16/17) and in Table S1. Please note that we had a small calculation error for the rate of embryonic lethality in our initial version of the manuscript. This has now been corrected. The conclusions made initially are not affected.

With respect to tumor phenotypes, we now show the tumor spectrum in different control cohorts, including *ITP2-M;Blm^{m3/m3}* and *Rosa26^{PB/+};Blm^{m3/m3}*. We show that the tumor spectrum is primarily dictated by the *Blm* background, although there are some “new” entities in the *ITP2-M;Rosa26^{PB/+};Blm^{m3/m3}*, which are not observed without transposon mutagenesis. In addition, transposon-driven mutagenesis accelerates *Blm*-related tumorigenesis.

The results are presented in Table S2 and described on page 6, 2nd paragraph of the manuscript.

5. With the major claim being that this is a new model for B cell lymphoma, the authors present very little data (only supplemental Figure 2 and supplemental table 1) that is convincing that they have a B cell lymphoma phenotype. The authors should elaborate on how their pathologists scored the tissues, what tissues they assessed, and if any additional histological markers were used. In addition, have the authors performed any flow cytometric analyses on the samples to better characterize the lymphomas? Was Igh or Igk rearrangement data obtained? Please provide more evidence to support this major claim.

As suggest by the reviewer, we now extensively characterized the B-cell lymphomas emerging from this screen at various levels:

Histopathology/Immunohistochemistry

We first performed a detailed histopathological classification of the lymphomas based on tumor morphology and marker expression (immunohistochemistry). All analyses were conducted by Leticia Quintanilla-Martinez, who has long-standing expertise in mouse hematopathology.

We characterized 59 hematopoietic tumors using an immunohistochemistry (IHC) panel comprising the markers B220 (specific for B cells), CD3 (T cells), myeloperoxidase (myeloid cells) and CD138 (plasma cells). The vast majority of tumors were B cell neoplasms (52/59; 88.1%). Only six CD3 positive T-cell lymphomas (10.2%) and one tumor with myeloid differentiation (1.7%) were found.

B-cell lymphomas were almost exclusively reminiscent of human diffuse large B-cell lymphoma (DLBCL; 51/52; 98.1%). Neoplasms usually manifested in mesenteric lymph nodes and/or spleens, with moderate or extensive alterations of lymphoid organ architecture due to abnormal B-cell expansion (shown by B220 IHC). DLBCLs were composed of large-sized neoplastic cells (centroblasts) with abundant cytoplasm, a round nucleus, vesicular chromatin, with two or more nucleoli, which were often membrane-bound, and showed high proliferation rates (as demonstrated by Ki-67 IHC).

In all tumors, we also observed a small percentage of lymphoid cells with immunoblastic morphology (larger cells with abundant cytoplasm and one prominent centrally localized nucleolus).

A subset of DLBCL cases showed characteristics of plasmacytic differentiation (13/51; 25.5%), in which a subset of tumor cells lost the B cell marker (B220) and expressed the plasma cell marker

(CD138). A significant proportion of tumor cells retained however B220 expression, distinguishing these cancers from plasmablastic lymphoma or other plasma cell malignancies. In general, tumor cell infiltration into organs located within the thoracic and abdominal cavities such as lungs, liver, intestine and kidneys was frequently observed (37/42 analyzed DLBCLs; 88.1%) while bone marrow infiltration was rare (2/42; 4.8%).

To sub-classify the DLBCL cases based on their cell of origin, we performed IHC using the germinal center marker Bcl6 and the post-germinal center marker Mum1/Irf4. Expression of MUM1/IRF4 is associated with non-GCB DLBCL in humans. In contrast, BCL6 expression is primarily associated with germinal center B-cell like- (GCB) DLBCL, although a subset of non-GCB DLBCL is also positive for BCL6. We analyzed 20 samples and found that 15 cases (75%) were Bcl6-positive/Irf4-negative, suggesting a GCB DLBCL phenotype. Five cases (25%) were Irf4-positive/Bcl6-negative, which we classified as non-GCB DLBCL.

The results are shown in Figure 2, Figure S3, Figure S4 and Table S4 and are referred to on pages 6 and 7 of the manuscript.

Gene expression profiling

In addition, we performed RNA sequencing of DLBCL samples ($n = 25$) for gene-expression profiling (GEP), which is considered the gold standard for DLBCL sub-classification in humans. Using the murine orthologues of the human classifier genes, mouse DLBCLs clustered into two main clusters. Cluster B contained exclusively IHC-diagnosed GCB tumors, whereas all five IHC-diagnosed non-GCB cancers fell into cluster A. Like in human DLBCL, IHC-based and GEP-based tumor classification are not fully superimposable. In fact, the discordance in mice might be even stronger because mouse DLBCLs are less “homogeneous”: we observed that GCB tumors often contain infiltrates of CD3 positive T lymphocytes and residual plasma cells, which makes accurate GEP from whole tumor lysates challenging. This might be a reason why some of the IHC-diagnosed GCB samples fall into cluster A. Taken together these data show mouse DLBCLs can be sub-classified similarly to human DLBCLs. Thus, our model recapitulates key aspects of the human disease.

The results can be found in Figure 2 and are referred to on page 7, 3rd paragraph of the manuscript.

Clonality

For clonality analysis, we performed RNA-based immunoglobulin repertoire profiling of 30 DLBCL cases. To this end, we conducted full-length amplification of the variable regions of the immunoglobulin heavy and light chains. To eliminate PCR and sequencing errors leading to incorrect clone assignments, unique molecular identifiers (UMI) were introduced during cDNA synthesis. Immune repertoires were sequenced on Illumina MiSeq. For data analysis, de-multiplexing and UMI consensus sequence assembly was performed with *MIGEC*. *MiXCR* was used for clone detection based on the highly variable complementarity-determining region 3 (CDR3). To visualize the clonal structure of tumors, we developed a custom script to generate

clonality network plots. In these plots, each clone constitutes a node of the network. The size of the node correlates with the number of reads assigned to it, and clones differing by only 1 bp in their CDR3 sequence are linked. The complexity of the branching (i.e. number of subclones) is a measure for the grade of somatic hypermutation (SHM), which is a hallmark of DLBCL. We highlighted clones that consist of more than 10% of the total reads of a sample in color (red, blue or green). Different clones (as defined by a unique V(D)J rearrangement) are marked with different colors. Note that RNA was isolated from whole tumor tissue lysates that contain varying amounts of non-transformed B cells (most likely accounting for the small gray nodes in the plots).

The vast majority of tumors (16/27) were monoclonal, indicating that these were full-blown malignant lymphomas arising from one transformed B cell. Eight samples consisted of two dominant clones, suggesting the presence of two independent malignant DLBCLs in one mouse. Only two tumors arose from multiple clones. We found evidence of SHM in the majority of tumors. As expected, there were differences in the extent of SHM between individual tumors, and between heavy and light chains of the same clone, as these undergo SHM separately.

The results are shown in Figure 3, Figure S5, Figure S6, Figure S7, Figure S8, Table S5 and are referred to on pages 7 and 8 of the manuscript.

6. Related to the issue of Ig gene rearrangement, is the disease presented oligoclonal or does each animal have a multiple lymphoid disease being driven by unique clones?

Please also see above (answer to question 5). The vast majority of tumors (16/27) were monoclonal, indicating that these were full-blown malignant lymphomas arising from one transformed B cell. Eight samples consisted of two dominant clones, suggesting the presence of two independent malignant DLBCLs in one mouse. Only two tumors arose from multiple clones. We found evidence of SHM in the majority of tumors. As expected, there were differences in the extent of SHM between heavy and light chains, which undergo SHM separately, as well as between individual tumors.

The results are shown in Figure 3, Figure S5, Figure S6, Figure S7, Figure S8, Table S5 and are referred to on pages 7 and 8 of the manuscript.

7. Related to the analysis of common insertion sites (CIS):

a. What about using additional methods to analyse the data to more comprehensively identify CIS genes (other statistical methods?)

To identify CIS genes with a second statistical approach, we now also performed TAPDANCE (Transposon Annotation Poisson Distribution Association Network Connectivity Environment) analysis. Although Tapdance deploys statistics that are more stringent, all top 50 CIS genes found

by CIMPL were also identified using TAPDANCE analysis, underlying the reliability and reproducibility of the CIS identification approaches.

The results are shown in Table S6 and referred to on page 9, 2nd paragraph and on page 21, 2nd paragraph (methods section) of the manuscript.

b. Are any co-occurring CIS observed?

As suggested by the reviewer, we performed co-occurrence analyses of the top 50 CIS genes using a Fisher's exact test. The data are shown in Table S8 and referred to on page 9, 2nd paragraph of the manuscript.

8. The authors comment that 29% of animals developed various solid cancers. What are they? Could some of these actually been lymphoid disease invading other tissues? A supplemental figure/table describing these observations is needed to have a fully comprehensive view of the model.

We included a comprehensive list of all solid tumors diagnosed in *IPB* mice in Table S3 and show histological images of five exemplary solid tumors in Figure S2. All solid tumors were characterized histopathologically as clear carcinomas (i.e. tumors of epithelial origin) excluding the possibility of being infiltrating lymphomas.

The data are referred to on page 6, 2nd paragraph of the manuscript.

9. The authors can refer to Moriarity et al. 2015 Nature Genetics to support their hypothesis about transposon-driven mutagenesis is a key cancer-promoting factor that can drive tumours in absence of chromosomal rearrangements. Similar observations were made in an osteosarcoma model using SB.

We now refer to Moriarity *et al.* Nature Genetics 2015 as a previous study identifying transposon-driven mutagenesis as a key cancer-promoting factor on page 9, 1st paragraph of the manuscript.

10. With regard to comparisons of mouse BCL CIS-associated genes to human DLBCL genomic alterations, the authors used gene expression levels but an analysis of gene copy number loss would be useful to include. That is, how many BCL CIS-associated genes show copy number loss or gain in human DLBCL. The authors speculate that some of the BCL CIS-associated genes are lost in human DLBCL rather than suffering obvious loss of function mutations.

As suggested, we now included a figure with oncoplots showing copy number alterations of candidate tumor suppressor genes in human DLBCL. The existence of copy number alterations in human orthologues of the 50 top CIS genes in our screen was interrogated in the TCGA-DLBC dataset (Pan-Cancer Atlas from Cancer Genome Atlas (TCGA) consortium). Featured are genes with CNV, as estimated by GISTIC 2.0 analysis. These analyses in human DLBCLs revealed almost exclusively deletions at loci harboring human orthologues of mouse CIS genes (Figure S12). These cross-species comparisons thus provide further support for the power of the screen to identify tumor suppressors regulated in human DLBCL.

The results are described on page 10, 2nd paragraph of the manuscript.

11. The Crispr/Cas9 based in vivo model presented is a powerful system to validate genetic drivers for many hematopoietic diseases. The authors need to provide more evidence (histology, flow) describing the FLC populations targeted for modification and the resulting phenotypes. No data are shown to support lymphoma phenotypes. Moreover, the lymphomas induced by loss of Phip and Rfx7 should be analysed at the level of protein, that is, do the lymphomas produced express reduced or no protein for the target gene.

As suggested by the reviewer, we now included data from flow cytometry analysis of fetal liver cells, which were transduced with lentiviral GFP-tagged sgRNA vectors for candidate gene knockout. Corresponding FACS plots are shown in Figure S16. DAPI staining was used to discriminate dead (DAPI positive) from viable (DAPI negative) cells. To define the fraction of transduced cells in the DAPI-negative gate, we determined the percentage of GFP-positive cells, which typically ranged between 20% and 30%.

Exemplary flow cytometry plots are shown in Figure S16 and the data are referred to on pages 12 and 13 of the manuscript.

For characterization of the hematopoietic tumors emerging from the validation model, we conducted H&E stainings to visualize tissue morphology/structure as well as B220 IHC to identify malignant B-cell expansions. Four different expert pathologists in mouse or human hematopathology independently diagnosed the cancers as malignant lymphomas of B-cell origin.

Exemplary microscopic images are shown in Figure S18 and the data are referred to on pages 12 and 13 of the manuscript.

To quantify Rfx7 and Phip expression in lymphomas, we first performed Western blotting. We tested all four commercially available antibodies extensively, but could only detect non-specific bands in our positive controls. Therefore, we next quantified the expression of Rfx7 and Phip by

real time quantitative PCR in bulk tumor tissue. The rationale for this experiment is that frameshift mutations can not only lead to premature termination of translation but can also induce nonsense mediated mRNA decay, in which case the mRNA amount would be expected to drop. We indeed found substantially reduced Rfx7 and Phip mRNA expression in Rfx7-sgRNA and Phip-sgRNA tumors, respectively. Given that these analyses were performed in bulk tumors that also contain non-tumor cells, the actual effect might be even stronger. Thus, altogether we provide at three levels evidence for the validity of the CRISPR approach to functionally analyze cancer genes: (i) the proof/detection of CRISPR-induced frame-shift mutations at the DNA level, (ii) the drop in mature RNA, (iii) the strong acceleration of tumorigenesis by CRISPR editing of respective target genes.

The results are shown in Figure S18 and are referred to on pages 12 and 13 of the manuscript.

12. When the author states “strikingly high number of genes involved in...” please put in the % of genes that fall into your descriptors and whether enrichment is statistically significant?

We changed this in the manuscript (page 12, 3rd paragraph). Of the top 50 gene list, 22 genes (44%) were identified as transcriptional (15 genes; 30%) or chromatin (7 genes; 14%) regulators. Sun *et al.* 2017 list 1603 known transcription factors in the mouse genome. Comparing this number to the 15 genes identified as transcription factors in top 50 gene list, the enrichment for transcription factors in our screen is highly significant ($p = 1.4 \times 10^{-5}$, Fisher’s exact test).

This information was added to the legend of Figure 4.

Suggested changes to figures:

- *Figure 1:*
 - *include the evidence that disease is really BCL*
We now included extensive evidence that the disease is indeed full-blown malignant DLBCL: shown in Figures 2 and 3 of the manuscript.
 - *include the double transgenic animals on survival curves*
All available experimental and control cohorts are included in Figure 1. Please see our response to comment 4.
 - *embryonic lethality bar graph doesn’t add much and can just be mentioned in text.*
We removed the bar graph from the figure, as suggested.

- *Figure 2: The cellular localization data is really glossed over in the text and therefore is not really necessary. If authors highlight specifically a gene with unknown function and make suggestions about it, then it would be worth keeping.*
As suggest by the reviewer, we removed the cellular localization data from the figure.

- *Figure 3: Could be lumped in with Figure 2, doesn't need to be stand alone.*

Figure 2 (new Figure 4) shows mouse copy number and transposon insertion data whereas and Figure 3 (new Figure 5) displays human expression data. Because the revised version of the manuscript contains a large amount of additional complex data and figures, we feel it is essential to not mix mouse and human data into individual figures, as it might further increase complexity and reduce readability of the revised paper.

- *Figure 4: Provide evidence that animals are developing BCL*

We now include microscopic images of the B-cell lymphomas resulting from the CRISPR/Cas based validation experiments. The results are shown in Figure S18.

- *Figure 5: Can be combined with figure 4 and then performing similar analysis on Rfx7 as well.*

As suggested by the reviewer, we combined figures 4 and 5 (new Figure 7). We also performed copy number analyses for *Rfx7*, as suggested. We found no recurrent aberrations at this locus in human DLBCL.

- *Figure 6:*

- *Although interesting, the figure itself is underwhelming. Perhaps focus on NR3C1 that is highlighted in the discussion with including more data on this particular pathway. Include data on statements made about mutations, etc....*

- *The other survival curves could be placed in a supplemental figure.*

We agree that parts of this figure could be moved to the supplement, as we do not perform in-depth discussions/analyses on all the individual genes listed in the figure. Two other reviewers emphasized however the importance of translational/clinical data. This figure provides an easy to interpret global view of cancer genes discovered in our screen with clinical relevance (reduced expression associated with poor patient survival). We therefore refrained from moving the figure to the supplement. As suggested, we now discuss the potential relevance of the *Nr3c1* pathway for DLBCL biology and comment that we our screen also discovered genes that are interacting with or downstream of *Nr3c1*, including the CIS genes *Crebbp*, *Pou2f1*, and *Hnrnpu*, further supporting the relevance of this pathway.

We included this information on page 14, 2nd paragraph of the manuscript.

REVIEWERS' COMMENTS:

Reviewer #2 (Remarks to the Author):

The authors addressed all my comments and concerns. I applaud the authors for this manuscript.

Reviewer #3 (Remarks to the Author):

The authors have made significant amendments to the initial manuscript which address all main points of concern raised by the reviewers. From my own perspective, questions as to the immunogenicity of Cas9 in the models used as well as the pathway-based analysis of CIS genes have been addressed and I am happy for the paper to proceed to publication.

Overall, this is a well-designed experiment that demonstrates the ability of genetic screens to compliment patient tumour sequencing in an understanding of those genes complicit in tumourigenesis, and will be of great interest to the readers of the journal.

Reviewer #4 (Remarks to the Author):

The revised manuscript by Weber et al. describes a PiggyBac transposon forward genetic screen for identifying genetic drivers of B cell lymphoma. It also includes a new in vivo validation approach of candidate B cell lymphoma driver genes. This manuscript presents a large body of work which is now far more complete. For example, more tumor RNA sequencing and comparisons with human B cell lymphomas were done. The models are much more well defined and the comparative oncogenomics with human B cell lymphomas is very complete. The data do reveal important new human B cell lymphoma tumor suppressor genes (TSG) and pathways and this is innovative and impactful work. It goes well beyond prior B cell lymphoma papers using transposon mutagenesis. An important outcome is the new observation in this revised work that loss of heterozygosity is enhanced at TSGs inactivated by transposon insertion on a Blm gene mutant background that really proves the utility of their overall approach. Also useful are new data on embryonic lethality based on transposon copy number on a Blm gene mutant background. Revisions to the text and figures also improve the readability and impact of the paper

Weber, de la Rosa, Grove et al. “*PiggyBac* transposon tools for recessive screening identify B-cell lymphoma drivers in mice”

Point by point responses to the reviewers' comments

Reviewer #2 (Remarks to the Author):

The authors addressed all my comments and concerns. I applaud the authors for this manuscript.

Reviewer #3 (Remarks to the Author):

The authors have made significant amendments to the initial manuscript which address all main points of concern raised by the reviewers. From my own perspective, questions as to the immunogenicity of Cas9 in the models used as well as the pathway-based analysis of CIS genes have been addressed and I am happy for the paper to proceed to publication.

Overall, this is a well-designed experiment that demonstrates the ability of genetic screens to compliment patient tumour sequencing in an understanding of those genes complicit in tumourigenesis, and will be of great interest to the readers of the journal.

Reviewer #4 (Remarks to the Author):

The revised manuscript by Weber et al. describes a PiggyBac transposon forward genetic screen for identifying genetic drivers of B cell lymphoma. It also includes a new in vivo validation approach of candidate B cell lymphoma driver genes. This manuscript presents a large body of work which is now far more complete. For example, more tumor RNA sequencing and comparisons with human B cell lymphomas were done. The models are much more well defined and the comparative oncogenomics with human B cell lymphomas is very complete. The data do reveal important new human B cell lymphoma tumor suppressor genes (TSG) and pathways and this is innovative and impactful work. It goes well beyond prior B cell lymphoma papers using transposon mutagenesis. An important outcome is the new observation in this revised work that loss of heterozygosity is enhanced at TSGs inactivated by transposon insertion on a Blm gene mutant background that really proves the utility of their overall approach. Also useful are new data on embryonic lethality based on transposon copy number on a Blm gene mutant background. Revisions to the text and figures also improve the readability and impact of the paper

We would like to thank the reviewers for their very positive comments. We are delighted that they find the manuscript suitable for publication in *Nature Communications*.